# Bayesian Neighborhood Adaptation for Graph Neural Networks

**Paribesh Regmi**                                                *pr8537@rit.edu*
*Golisano College of Computing and Information Science*
*Rochester Institute of Technology*

**Rui Li**[*]                                                      *rxlics@rit.edu*
*Golisano College of Computing and Information Science*
*Rochester Institute of Technology*

**Kishan KC**                                                     *ckshan@amazon.com*
*Amazon.com, Inc.*

**Reviewed on OpenReview:** *https://openreview.net/forum?id=2zEemRib3a*

## Abstract

The neighborhood scope (i.e., number of hops) where graph neural networks (GNNs) aggregate information to characterize a node's statistical property is critical to GNNs' performance. Two-stage approaches, training and validating GNNs for every pre-specified neighborhood scope to search for the best setting, is a time-consuming task and tends to be biased due to the search space design. How to adaptively determine proper neighborhood scopes for the aggregation process for both homophilic and heterophilic graphs remains largely unexplored. We thus propose to model the GNNs' message-passing behavior on a graph as a stochastic process by treating the number of hops as a beta process. This Bayesian framework allows us to infer the most plausible neighborhood scope for message aggregation simultaneously with the optimization of GNN parameters. Our theoretical analysis shows that the scope inference improves the expressivity of a GNN. Experiments on benchmark homophilic and heterophilic datasets show that the proposed method is compatible with state-of-the-art GNN variants, achieving competitive or superior performance on the node classification task, and providing well-calibrated predictions.

## 1 Introduction

Graph neural networks (GNNs) (Kipf & Welling, 2016) and its variants have shown success in modeling graph-structured data arising in various fields, such as computational biology (Huang et al., 2020; Kishan et al., 2021), social information analysis (Li & Goldwasser, 2019; Qiu et al., 2018), recommender systems (Ying et al., 2018), etc. Due to the locality assumption, multiple GNN layers needed to be stacked up in the network structures in order to expand the neighborhood scope for message aggregation. Substantial research efforts focus on enhancing the aggregation schemes for homophilic and heterophilic graphs, resulting in GNN variants showing significant performance improvements (Xu et al., 2018; Veličković et al., 2017; Rong et al., 2020; Chen et al., 2020; Chien et al., 2021; Luan et al., 2022; Zeng et al., 2021). Although the neighborhood scope where GNNs aggregate information is also vital to their performance, the state-of-the-art GNN variants still rely on traditional two-stage approaches to search for the best setting. Since these empirical approaches involve training and validating GNN models for each single candidate configuration of neighborhood scope, it is a daunting task and tend to be biased. Moreover, since the validation error is a noisy quantity, it is necessary to devote large quantities of data to the validation set to obtain a reasonable signal-to-noise ratio.

---

[*]Corresponding author

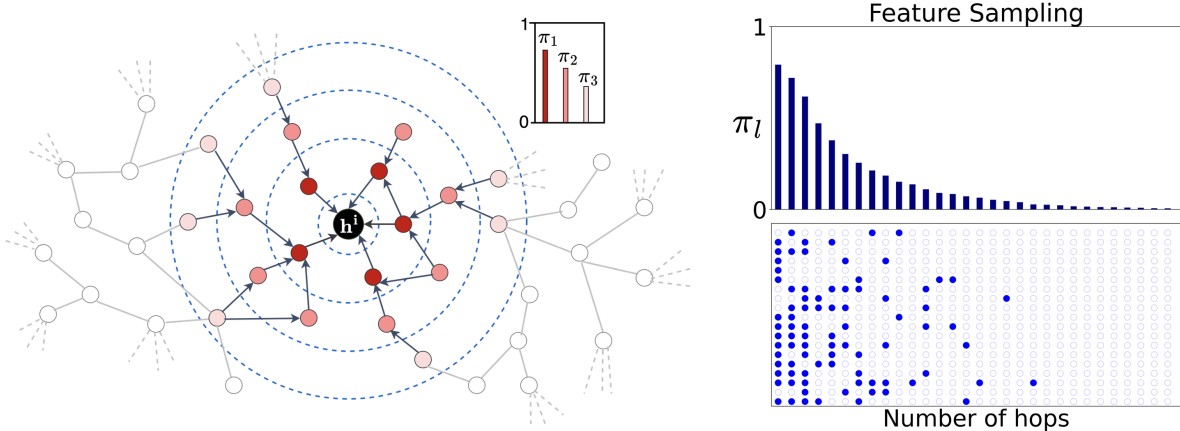

Figure 1: Illustration of our proposed neighborhood adaptation strategy. **Left:** The feature of a given node (black-colored) is generated by aggregating messages from neighbors located multiple hops away. The direction of message passing is indicated by arrows. The nodes in each hop $l$ are assigned a contribution probability ($\pi_l$) indicating their contribution in aggregation (color-coded). **Right:** Visualizing stick-breaking construction of a beta process. The sticks on top are random draws from a beta process, representing the probabilities over the number of hops. The bottom shows the conjugate Bernoulli process over node feature dimensions. Filled circles (blue) indicate a random draw of 1 confirming the selection of a particular feature.

Recent research efforts mainly focus on designing aggregation schemes for effective message passing to improve GNNs' performance. Regularization-based methods (Rong et al., 2020; Hasanzadeh et al., 2020), introduce regularization techniques that randomly drop edges or neural connections between layers during training. Connection-based methods (Xu et al., 2018; Chen et al., 2020) incorporate additional residual connections between GNN layers. Another group of methods (Abu-El-Haija et al., 2019; Wu et al., 2019) aggregate messages from multiple hops in a single neural layer by using higher powers of the adjacency matrix. GAT (Veličković et al., 2017) enables the prioritization of specific nodes during message aggregation in a pre-specified neighborhood scope. The performance of some approaches rely on an implicit assumption of graph homophily (McPherson et al., 2001) (i.e., nodes belonging to the same class tend to form edges) and they may not perform well on heterophilic graphs (i.e., nodes with distinct features are more likely connected) (Zhu et al., 2020; Liu et al., 2021). Aggregation schemes (Chien et al., 2021; Luan et al., 2022; Zhu et al., 2020) tailored for heterophilic settings allow GNN variants to achieve state-of-the-art performance. Besides effective aggregation schemes, proper neighborhood scopes for message passing is also critical for GNNs' superior performance (Huang et al., 2020; Abu-El-Haija et al., 2019; Perozzi et al., 2014). Small neighborhood scopes limit GNNs at capturing long-range information in the graph, whereas overly large neighborhood scopes tend to degrade model expressivity and incur expensive computation. It remains an open question how to automatically determine proper neighborhood scope for both homophilic and heterophilic graphs without numerous rounds of training and validating different GNN candidate structures.

To address this challenge, we propose a neighborhood scope adaptation strategy based on non-parametric Bayesian inference. This general framework allows us to infer the most plausible neighborhood scope for message aggregation simultaneously with learning node representations. Specifically, we model the expansion of the neighborhood as a stochastic process by defining a beta process prior over the number of hops. The beta process induces a probability for the neighboring nodes in each hop to quantify their contribution to the aggregation. Based on the hop-wise probabilities, we randomly sample a fraction of the node features by masking them with a binary vector generated from a conjugate Bernoulli process. Such a strategy further prioritizes the nodes' contribution within the neighborhood scope, leading to customized message aggregation. To assess the effectiveness of our proposed framework, we showcase its versatility on state-of-the-art GCN variants and demonstrate its ability to boost their performance on both homophilic and heterophilic datasets. We also provide theoretical and empirical analysis of its ability to improve expressivity in deep network structures. Moreover, our framework leads to well-calibrated predictions via reliable uncertainty

estimation. We also demonstrate that our method can successfully infer the L3 path (Kovács et al., 2019) of a real-world biomolecular network.

Our contributions are as follows: i) We propose a general Bayesian inference strategy that automatically determines neighborhood scopes for message passing. ii) We introduce an efficient stochastic variational approximation to simultaneously infer neighborhood scopes and learn node representations. iii) Our theoretical and empirical analyses show that our framework can enhance GNNs' expressivity. iv) We demonstrate the adaptive neighborhood scope inference boosts state-of-the-art GNN performance on node classification task for both homophilic and heterophilic graphs.

## 2 Preliminaries and Related Works

We denote a graph with $\mathcal{G}$ with vertices (nodes), edges, and node features denoted by $(\mathcal{V}, \mathcal{E}, \mathbf{X})$. The adjacency matrix with added self-connections is denoted by $\mathbf{A} \in \mathbb{R}^{|\mathcal{V}| \times |\mathcal{V}|}$ and $\widehat{\mathbf{A}}$ is it's normalized form. $\mathbf{H}_l$ denotes the $l^{th}$ hidden layer in a neural network ($\mathbf{H}_0 = \mathbf{X}$), with $\mathbf{W}_l$ being its parameters. GCN (Kipf & Welling, 2016) proposed a neural network with graph convolution layers. A GNN with a single hidden layer is represented as:

$$Y = \sigma(\widehat{\mathbf{A}}\ \sigma(\widehat{\mathbf{A}}\mathbf{X}\mathbf{W}_0)\mathbf{W}_1) \tag{1}$$

where $\sigma$ is the activation function. Multiplication by adjacency matrix $\widehat{\mathbf{A}}$ denotes message aggregation from the immediate neighborhood. Eqn. (1) highlights that stacking multiple layers in a GNN model involves repeated multiplication with $\widehat{\mathbf{A}}$, expanding the neighborhood scope with each additional layer. Therefore, specifying the number of layers implicitly assumes the locality i.e. it incorporates $l^{th}$-hop neighbors for message aggregation when the network consists of $l$ layers. Subsequent research has aimed to improve upon this basic aggregation scheme, as outlined below.

### 2.1 Message Aggregation Schemes

Dropedge (Rong et al., 2020) proposes to randomly drop a fraction of the edges and train GNNs with the resulting sparse graph to reduce noise in the graph structure. Residual connections between layers are employed to enhance GNN models' performance while preserving locality. JKNet (Xu et al., 2018) aggregates the information from all hidden layers before feeding it into the output layer. Such aggregation helps maintain the local information of each layer when propagating towards the output layer. PPNP (Gasteiger et al., 2019) proposes a message aggregation scheme based on the personalized PageRank algorithm (Page et al., 1998), allowing message passing from larger neighborhood. GCNII (Chen et al., 2020) extends GCNs with an initial residual connection and identity mapping, resulting in stable and better performance with deeper structures. In addition, utilizing higher powers of the adjacency matrix for aggregation from broader neighborhoods is also an effective strategy. The GNN variant proposed in (Wu et al., 2019) widens the scope from immediate neighbors to the ones lying multiple hops away in a single layer and effectively expand the neighborhood scope for aggregation. Mixhop, (Abu-El-Haija et al., 2019), employs multiple heads in a single layer to aggregate and combine messages from neighbors lying at higher hops. Co-GNN Finkelshtein et al. (2024) proposes a flexible message passing approach where each nodes can determine how the message is propagated to and from its neighbors. At each layer, the message passing behavior of each nodes can change, which results in a dynamic, task-aware computational graph that is different at each layer. D2GCN Duan et al. (2022) incorporates a subset of non-neighboring nodes as negative samples during message passing and learns the central node's representation to deviate away from the nodes. The extends the message passing scheme to non-neighboring nodes as well. LDGCN Duan et al. (2024) proposes diverse negative sampling that improves on D2GCN by reducing the redundancy of negative samples at each layer, which is also shown to improve the expressivity of GCNs. All these methods lack an automatic mechanism to determine the appropriate neighborhood scope from the input graph. Attention-based methods like GAT (Veličković et al., 2017) learn edge-specific attention weights to prioritize certain neighbors during message aggregation. However, this prioritization is limited to a fixed neighborhood scope determined by the number of layers. Reinforcement learning approaches such as Policy-GNN Lai et al. (2020) adaptively select the number of hops for each

node using a deep Q-network, but this requires training an additional network. Furthermore, the maximum aggregation range ($K$) must still be predefined, and the adaptivity is confined within this range. In contrast, our method requires no auxiliary network and imposes no fixed upper limit on the neighborhood scope, allowing information to propagate theoretically up to an infinite range.

## 2.2 Aggregation Schemes for Heterophilic Graphs

Some aggregation approaches assume graph homophily and perform poorly on heterophilic graphs (Pei et al., 2020; Bojchevski et al., 2020; 2019). Since the connected nodes often exhibit significantly different properties in heterophilic graphs, a new design of effective message aggregation becomes necessary (Jia & Benson, 2020). To address this challenge, $H_2$GCN (Zhu et al., 2020) proposes ego embedding and higher order neighborhood aggregation, allowing for significant performance improvements on heterophilic graphs. GPR-GNN (Chien et al., 2021) associates message aggregation in each step with a learnable weight, allowing it to adapt to the homophily or heterophily structure of the input graph. ACM-GCN (Luan et al., 2022) proposes adaptive channel mixing and achieves state-of-the-art results on benchmark heterophilic datasets. However, all these methods rely on pre-specified neighborhood scope prior to training (the number of propagation steps in $H_2$GCN and GPR-GNN or the number of graph convolutional layers in ACM-GCN) the input graph. Moreover, while all the mentioned methods are

## 2.3 Bayesian Methods for GNNs

Bayesian approaches have been applied to graph modeling, offering key advantages such as strong performance with limited labeled data, robustness to adversarial perturbations, and reliable uncertainty calibration. These are helpful in scenarios such as safety-critical systems, few-shot learning, and adversarial settings, where improved robustness and reliable uncertainty estimation can lead to more reliable and cautious decision-making. A set of Bayesian approaches define a prior over the properties of the input graph, such as node features, structure, and labels, and perform joint inference to predict the labels of missing nodes in an unsupervised setting. Bayesian-GCNN (Zhang et al., 2019) considers the input graph as a specific realization from a parametric family of random graphs and performs inference of the joint posterior of the random graph parameters and the node labels. $G^3$NN (Ma et al., 2019) defines a random graph model where the distribution of random graphs also depends on the node features and labels, capturing their interactions for more flexible modeling, and infers missing labels in a semi-supervised learning setting. VGCN (Elinas et al., 2020) defines a probability distribution over the adjacency matrix to capture the topological structures of input graphs, enhancing model performance under adversarial perturbations of the input graph structure. Another approach, BBGDC (Hasanzadeh et al., 2020), which models both graph properties (edges) and GCN properties (convolutional channels) using a Bernoulli process. Both DropEdge and dropout (Srivastava et al., 2014) can be viewed as special cases of BBGDC. All these methods suffer from the limitation of relying on a GCN with a predefined neighborhood scope. Our method differs fundamentally from these approaches. While they define priors and conduct inference over the properties of input graphs, such as features, structure, and labels, we propose a general neighborhood scope inference strategy by defining a prior over neighborhood hops as and automatically inferring the neighborhood scope simultaneously with learning network parameters. Furthermore, the priors used in these approaches to model graph properties are not suitable for modeling the neighborhood scope of a GCN. This calls for a novel treatment of the scope, which we achieve through a stick-breaking construction of the beta process.

## 3 Bayesian Neighborhood Adaptation for GNNs

Given the importance of determining appropriate neighborhood scope for message aggregation, we propose a framework in which the model jointly learns the relevant neighborhood scope with the node representations. Specifically, we aim to learn discrete hop-level values that quantify the influence of neighbors at varying distances from a central node. Moreover, motivated by the robustness and reliable uncertainty calibration offered by Bayesian methods in graph learning (Elinas et al., 2020; Zhang et al., 2019; Hasanzadeh et al., 2020) (as discussed in Section 2.3), we adopt a Bayesian approach to infer the number of hops used for

message aggregation. In the following sections, we discuss the construction of the prior, define the likelihood model, and the posterior inference of our proposed Bayesian neighborhood adaptation framework.

### 3.1   Beta Process Prior over Infinite Neighborhood Scopes

We model the number of hops for message aggregation as a beta process (Paisley et al., 2010; Broderick et al., 2012). Beta process is a completely random measure defined over discrete indices that assigns each hop a contribution probability independently ranging from 0 to 1. Unlike other stochastic processes such as the Dirichlet or Pitman-Yor processes, which constrain probabilities to sum to one, the Beta process provides greater flexibility, making it a natural prior for our purpose. Specifically, we utilize stick-breaking construction of the beta process as follows:

$$\pi_l = \prod_{j=1}^{l} \nu_j, \quad \nu_l \sim \text{Beta}(\alpha, \beta) \tag{2}$$

where $\nu_l$ are sequentially drawn from a beta distribution. Additionally, $\pi_l$ denotes the contribution probability assigned to neighbors at the $l$-th hop level. Theoretically, the process assigns a probability to neighbors at infinite hop levels, potentially enabling message aggregation from an infinite scope, as demonstrated in Figure 1. $\pi_l$ can be interpreted as the contribution of nodes lying at the $l$-th hop during message aggregation. To sample features from nodes at the $l$-th hop, we introduce a Bernoulli variable $z_{ol} \sim \text{Bernoulli}(\pi_l)$. Thus, if $z_{ol} = 1$, it indicates that the $o$-th feature of a node at the $l$-hop level will be included for message aggregation. This mechanism selectively retains features from each hop based on $\pi_l$, allowing hops with higher contribution probabilities to have a stronger influence during aggregation, while those with lower probabilities contribute less, reflecting their weaker presence. We thus perform joint inference over the contribution probabilities of hops with a beta process and feature sampling using its conjugate Bernoulli process (KC et al., 2021; Regmi & Li, 2023; Thapa & Li, 2024) by formulating the prior over $\mathbf{Z}$ as:

$$p(\mathbf{Z}, \boldsymbol{\nu}|\alpha, \beta) = p(\boldsymbol{\nu}|\alpha, \beta)p(\mathbf{Z}|\boldsymbol{\nu}) = \prod_{l=1}^{\infty} \text{Beta}(\nu_l|\alpha, \beta) \prod_{o=1}^{O} \text{Bernoulli}(z_{ol}|\pi_l) \tag{3}$$

where $\alpha$ and $\beta$ are hyperparameters. Specifically, large $\alpha$ and small $\beta$ encourage aggregation from a broader neighborhood. Given the prior, next, we define a likelihood model.

### 3.2   GNN models as a Likelihood

As a likelihood model, we use a GNN backbone, in which we incorporate the feature sampling mechanism. Specifically, since GNNs aggregate messages from $l$-th hop neighbors at the $l$-th layer, we sample features of a node at the $l$-th hop level by multiplying the output from the $l$-th layer with the binary mask $\mathbf{z}_l$. Following this framework, a GNN layer is specified as:

$$\mathbf{H}_l = \sigma(\hat{\mathbf{A}}\mathbf{H}_{l-1}\mathbf{W}_l) \bigotimes \mathbf{z}_l + \mathbf{H}_{l-1}, \quad l \in \{1, 2, \ldots \infty\} \tag{4}$$

where $\mathbf{W}_l \in \mathbb{R}^{O \times O}$ denotes the weight matrix of layer $l$, $O$ is the dimensionality of the feature vector (i.e. the number of neurons in a hidden layer), and $\sigma$ is the activation function. The output of layer $l$ is multiplied element-wisely by a binary vector $\mathbf{z}_l$ where its element $z_{ol} \in \{0, 1\}$. The residual connections feed the outputs from the last activated GNN layer to the output layer. What's more, they also improve GNNs' performance with deep structures (Kipf & Welling, 2016).

Let $D = \{\mathbf{X}, \mathbf{Y}\}$ where $\mathbf{Y} = \{y_n\}$ denoting the node labels in a graph $\mathcal{G}$ with feature matrix $\mathbf{X}$. For the node classification task, we express the likelihood as:

$$p(D|\mathbf{Z}, \mathbf{W}, \mathcal{G}) = \prod_{n=1}^{|\mathcal{V}|} p(y_n|\hat{\mathbf{y}}_n), \quad \hat{\mathbf{Y}} = f(\mathbf{H}_L) \tag{5}$$

where $f(\cdot)$ denotes the output layer with softmax activation and $\widehat{\mathbf{Y}} = \{\hat{\mathbf{y}}_n\}$ is the estimated outputs. $\mathbf{Z}$ is a binary matrix whose $l$-th column is $\mathbf{z}_l$, $\mathbf{W}$ denotes the set of weight matrices, and $\mathbf{H}_L$ is the output from the last activated layer. The marginal likelihood obtained by marginalizing out $\mathbf{Z}$ in the product of Eqn. (3) and Eqn. (5) is:

$$p(D|\mathbf{W}, \mathcal{G}, \alpha, \beta) = \int p(D|\mathbf{Z}, \mathbf{W}, \mathcal{G})p(\mathbf{Z}, \boldsymbol{\nu}|\alpha, \beta)d\mathbf{Z}d\boldsymbol{\nu} \tag{6}$$

Given the likelihood and prior, we want to infer the posterior of $\mathbf{Z}, \boldsymbol{\nu}$. However, due to the non-linearity of GNN in the likelihood $P(D|\mathbf{Z}, \mathbf{W}, \mathcal{G})$ and $L \to \infty$ (Eqn. (3)), exact computation of the marginal likelihood in Eqn. (6), and thus the posterior is intractable. Therefore, we resort to variational inference to learn an approximate distribution.

### 3.3 Efficient Variational Approximation

We propose to adapt stochastic variational inference (Hoffman et al., 2013; Hoffman & Blei, 2015)) to approximate the marginal likelihood. We define a variational distribution as follows:

$$q(\mathbf{Z}, \boldsymbol{\nu}|\{a_l\}_{t=l}^T, \{b_l\}_{l=1}^T) = q(\boldsymbol{\nu})q(\mathbf{Z}|\boldsymbol{\nu}) = \prod_{l=1}^T \text{Beta}(\nu_t|a_l, b_l) \prod_{o=1}^O \text{ConBer}(z_{ol}|\pi_l) \tag{7}$$

where, $T$ is a truncation level that denotes the maximum number of layers for the variational distribution, $\text{ConBer}(z_{ol}|\pi_l)$ denotes a concrete Bernoulli distribution (Maddison et al., 2016; Jang et al., 2016). The concrete Bernoulli presents a continuous relaxation of the binary variables generated from the Bernoulli process. This relaxation allows optimization through gradient descent, which was not possible with the binary variables.[1] The neighborhood scope is defined as the index of the deepest activated layer i.e. layer with at least one activated neuron. Formally, the neighborhood scope $l^{ns}$ is the maximum value of $l$ such that $\sum_{o=1}^O z_{ol} > 0$.

The lower bound for the log marginal likelihood in Eqn. (6) (ELBO) is:

$$\log p(D|\mathbf{W}, \mathcal{G}, \alpha, \beta) \geq \mathbb{E}_{q(\mathbf{Z}, \nu)]}[\log p(D|\mathbf{Z}, \mathbf{W})] - \text{KL}[q(\boldsymbol{\nu})||p(\boldsymbol{\nu})] - \text{KL}[q(\mathbf{Z}|\boldsymbol{\nu})||p(\mathbf{Z}|\boldsymbol{\nu})] \tag{8}$$

Eqn. (8) is the optimization objective for our proposed framework. The first term in the right-hand side fits the model to the data and the rest two terms are regularization derived from the prior. The expectation is estimated with Monte Carlo sampling. The algorithm and diagram of our method are provided in Appendix D and E respectively.

## 4 Expressivity Analysis

For ease of notation, we use $N$ to denote the number of nodes in the graph ($N = |\mathcal{V}|$). For a symmetric adjacency matrix $\mathbf{A}$, the eigenvectors are perpendicular. Let $\lambda_1, \ldots, \lambda_N$ are the eigenvalues of $\mathbf{A}$ sorted in ascending order, and let the multiplicity of largest eigenvalue $\lambda_N$ is M i.e $\lambda_1 < \ldots, \lambda_{N-M} < \lambda_{N-M+1} = \cdots = \lambda_N$. We also assume that the adjacency matrix is normalized and possesses positive eigenvalues, with the maximum eigenvalue capped at 1.

Let, $\{e_m\}_{m=N-M+1,\ldots,N}$ be the orthonormal basis of the subspace $U$ corresponding to the eigenvalues $\{\lambda_m\}_{m=N-M+1,\ldots,N} = 1$, and $\{e_m\}_{m=1,\ldots,N-M}$ be the orthonormal basis for $U^\perp$. Consequently, $\mathbf{H} \in \mathbb{R}^{N \times O}$ can be expressed as $\mathbf{H} = \sum_{m=1}^N e_m \otimes w_m$ where $w_m \in \mathbb{R}^O$.

**Theorem 1** *(Oono & Suzuki, 2019) Let $d_{\mathcal{M}}(\mathbf{H})$ denote the perpendicular distance between the representations $\mathbf{H}$ and the subspace $U$, then the output representations from $L^{th}$ layer ($\mathbf{H}_L$) in a GCN exponentially converges to the subspace $U$.*

$$d_{\mathcal{M}}(\mathbf{H}_L) \leq \lambda^L d_{\mathcal{M}}(\mathbf{H}_0); \quad d_{\mathcal{M}}(\mathbf{H}) = \min_{\mathbf{P} \in U} ||\mathbf{H} - \mathbf{P}|| \tag{9}$$

$$\lambda = \max_{m \in \{1,..,N-M\}} \lambda_m$$

---

[1]Details on the concrete Bernoulli distribution is provided in Appendix J.

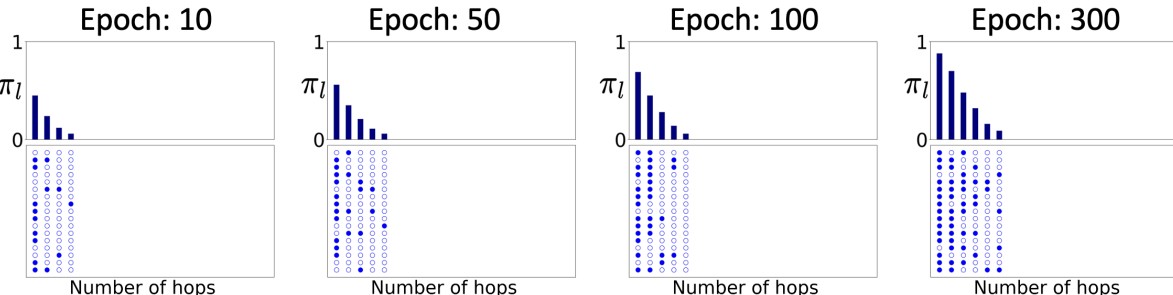

Figure 3: Evolution of neighborhood scope and contribution probabilities over the number of epochs for the Pubmed dataset when trained with our method. The contribution probabilities $\pi_l$ and hence the neighborhood scope increases as the training progresses and settles to an optimal value.

The convergence of $\mathbf{H}_L$ to lower dimensional subspace in Theorem 1 shows that the expressivity of a GCN exponentially decreases with an increase in the number of layers. This leads to information loss since nodes that lie within the same connected component tend to share identical features, making them indistinguishable.

In our analysis, we measure the convergence in terms of the angular region ($\theta$) spanned by $\mathbf{H}$ around $U$ as shown in Figure 2. A low value of $\theta$ is indicative of feature vector collapsing onto the lower-dimensional subspace $U$, resulting in decreased expressivity. Conversely, a high value of $\theta$ indicates that the feature representations span a broader angular region around $U$. The findings in (Kipf & Welling, 2016) indicate that incorporating residual connections in a GCN (ResGCN) yields improved performance with deeper structure compared to a vanilla GCN. We will theoretically analyze ResGCN and show that the addition of a residual connection widens the angular region $\theta$. Moreover, we observe that even in ResGCN, the region becomes more confined as the number of layers increases. Our proposed framework addresses this issue and maintains a wider region even with deeper layers by automatically inferring the relevant neighborhood scope. If $\theta_L$ is the angle spanned by $\mathbf{H}_L$ with the subspace $U$,

$$\tan \theta_L = \frac{d_{\mathcal{M}}(\mathbf{H}_L)}{|\mathbf{P}_L|}, \quad \mathbf{P} = \arg\min_{\mathbf{P} \in U} ||\mathbf{H} - \mathbf{P}||, \quad (10)$$

$$\theta_L = \tan^{-1}\left[\frac{d_{\mathcal{M}}(\mathbf{H}_L)}{|\mathbf{P}_L|}\right] \quad (11)$$

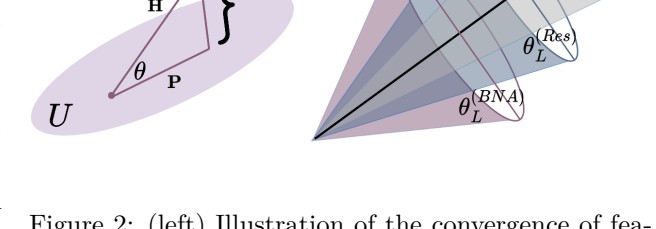

A network with residual connections between each layer (ResGCN) is represented as:

$$\mathbf{H}_l^{(Res)} = f_l(\mathbf{H}_{l-1}^{(Res)}) + \mathbf{H}_{l-1}^{(Res)} \quad (12)$$

**Lemma 1** *If $\theta_l^{(Res)}$ is the angle spanned by $\mathbf{H}_l^{(Res)}$ with the subspace $U$, then $\theta_L^{(Res)} \geq \theta_L$*

**Corollary 1** *The angular region narrows down with increase in layers L: $\theta_{L-1}^{(Res)} \geq \theta_L^{(Res)}$*

Figure 2: (left) Illustration of the convergence of feature vector $\mathbf{H}$ in the subspace $U$. $\mathbf{P}$ and $d_{\mathcal{M}}(\mathbf{H})$ are the projection and the perpendicular distance of $\mathbf{H}$ from the subspace respectively. $\theta$ is the size of the angular region spanned by $\mathbf{H}$ around $U$. (right) Visualization of the angular regions spanned by vanilla GCN (grey), ResGCN (blue), and BNA-GCN (purple) around the subspace $U$ (denoted by the dark line).

Proofs of the Lemma 1 and Corollary 1 are provided in Appendix B and B.1 respectively. The lemma suggests that ResGCN has better expressivity than a vanilla GCN with representations covering a wider angular region. However, Corollary 1 suggests that with the increase in layers in ResGCN, the features $\mathbf{H}_L^{(Res)}$ increasingly collapse into the subspace $U$.

Next, we analyze the impact of the application of our neighborhood inference framework in a GCN.

Table 1: Node classification performance of the GNN variants on homophilic (Cora, Citeseer, Pubmed) and heterophilic (Chameleon, Cornell, Texas, Wisconsin) graphs. The reported metric is averaged accuracy with ± one standard deviation. The best result for each graph is highlighted in bold, while the second-best result is underlined. (OOM indicates Out Of Memory)

| Baselines | Cora | Citeseer | Pubmed | Chameleon | Cornell | Texas | Wisconsin |
|---|---|---|---|---|---|---|---|
| GCN | 85.35±0.23 | 77.37±0.33 | 77.30±0.20 | 55.84±6.20 | 80.82±3.60 | 73.11±22.46 | 60.38±13.88 |
| ResGCN | 86.15±0.15 | 78.15±0.30 | 77.66±0.84 | 66.04±2.09 | 80.82±3.60 | 80.82±3.60 | 62.75±6.68 |
| BBGDC | 86.80±0.10 | 77.25±0.32 | OOM | 55.71±1.86 | 79.34±5.50 | 62.62 ± 8.08 | 59.88 ± 14.54 |
| Ours+ResGCN | 86.83±0.13 | 77.90±0.37 | 78.20±0.29 | 64.25±2.30 | 80.82±3.60 | 78.20±4.15 | 66.13±5.35 |
| JKNet | 86.20±0.10 | 77.68±0.35 | 77.25±0.23 | 52.43±2.84 | 74.43±8.30 | 70.00±5.49 | 62.13±6.20 |
| Ours+JKNet | 86.40±0.68 | 78.20±0.65 | 78.42±0.13 | 63.85±2.04 | 77.70±4.10 | 78.52±6.24 | 67.38±4.82 |
| GAT | 86.43±0.40 | 77.76±0.24 | 76.78±0.63 | 63.90±0.46 | 76.00±1.01 | 78.87±0.86 | 71.01±4.66 |
| Ours+GAT | 86.70±0.40 | 77.52±0.32 | 77.47±0.19 | 65.32±2.61 | 79.84±3.36 | 75.08±7.10 | 73.12±3.41 |
| GCNII | **87.53±0.30** | 77.63±0.21 | **79.96±0.17** | 58.97±2.76 | 87.70±5.15 | 76.07±5.35 | 80.37±5.86 |
| Ours+GCNII | 87.26±0.25 | **78.36±0.66** | 78.60±0.60 | 57.44±3.35 | 87.87±5.19 | 90.66±2.54 | 90.75±3.12 |
| GPR-GCN | - | - | - | 67.48±0.40 | 91.36±0.70 | 92.92±0.61 | 93.75±2.37 |
| ACM-GCN+ | 85.63±0.13 | 75.20±0.29 | 75.73±0.40 | **74.62±1.79** | 92.46±2.34 | 91.80±4.21 | 94.87±2.20 |
| Ours+ACM-GCN+ | 84.76±0.76 | 74.73±0.33 | 74.33±0.82 | 74.38±1.69 | **93.61±2.13** | **94.10±3.53** | **95.75±1.79** |

**Theorem 2** *With the application of the Bayesian Neighborhood Adaptation (BNA) framework, if $\theta_L^{(BNA)}$ is the angle spanned with the subspace $U$, then $\theta_L^{(BNA)} \geq \theta_L^{(Res)} \geq \theta_L$.*

**Corollary 2** *Beyond a certain number of layers $l^{ns}$, the angular region $\theta_L^{(BNA)}$ remains constant even with further increase in the number of layers: $\theta_L^{(BNA)} = \theta_{l^{ns}}^{(BNA)}$ for $L \geq l^{ns}$*

Proofs of Theorem 2 and Corollary 2 are provided in Appendix C and C.1 respectively. The theorem suggests that the application of the BNA framework further enhances the expressivity of ResGCNs. Importantly, Corollary 2 suggests that by inferring the appropriate neighborhood for message aggregation, the BNA framework avoids feature collapse and prevents information loss in a deep GCN. These analyses are illustrated in Figure 2 and empirically validated in Figure 5.

## 5 Experiments

We present a series of experiments to assess the effectiveness of the proposed neighborhood scope inference framework. In section 5.1, we demonstrate the neighborhood scope inference mechanism by showing how the scope of a GNN expands during training. In section 5.2, we then compare the performance of GNN baselines with and without the application of our framework. In section 5.3, we empirically validate the expressivity analysis from section 4 by visualizing the learned node representations of GCN and our method. In section 5.4, we assess the effect of our framework on uncertainty calibration by analyzing the uncertainty estimates. We then conduct an ablation study in section 5.5 to assess the contribution of each component. In Section 5.6, we evaluate the scalability of our framework to large graph datasets. Section 5.7 provides both theoretical and empirical analysis of the time and space complexity of our method. Lastly, in section 5.8, we present a case study on a real-world molecular graph dataset to validate the neighborhood scope inference capability of our approach.

### 5.1 Neighborhood Scope Adaptation

Figure 3 demonstrates the mechanism of neighborhood scope adaptation during training over 300 epochs for the Pubmed dataset. The truncation was set to $T = 10$. During initial phase, the scope is limited to 4 hops, with comparatively less contribution from each hop. As the training progresses, our framework allows the contribution probabilities and hence the neighborhood scope to adapt to the input. At $300^{th}$ epoch, the expansion converges to 6 hops, and the contribution probabilities become stable over training.

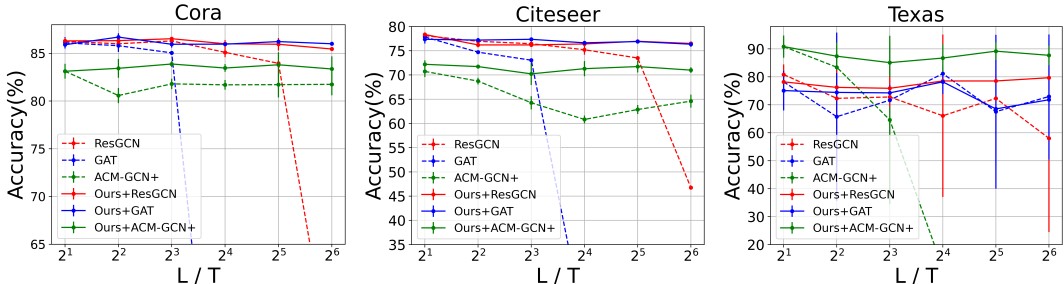

Figure 4: The impact of increasing the depths ($L/T$) of GNN variants with and without our framework on their expressivity. Although the depth increase degrades the performance of vanilla ResGCN, GAT, and ACM-GCN, the application of our framework stabilizes their performance even for deep network structures.

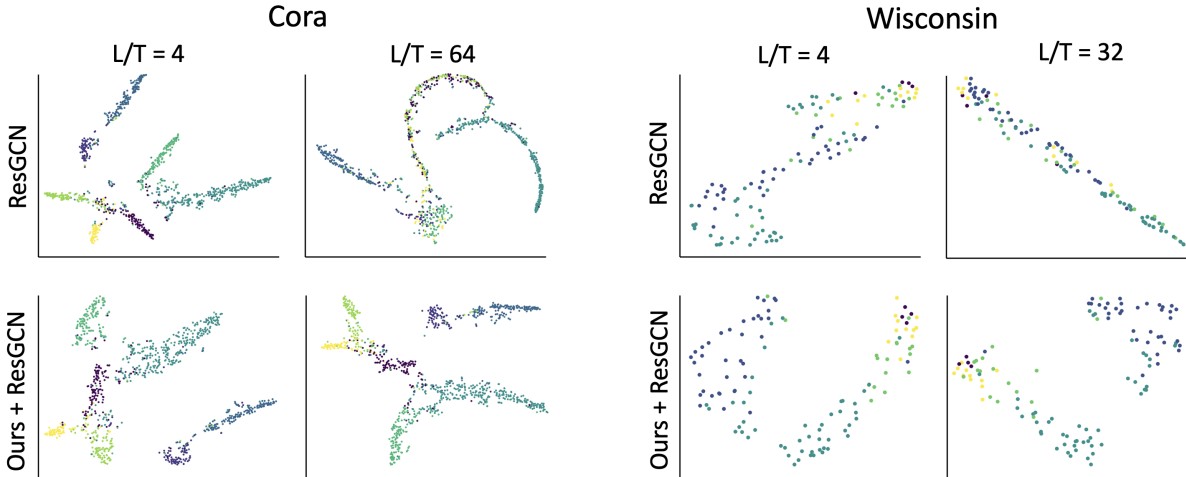

Figure 5: TSNE visualization of the learned node representations by ResGCN with and without our framework for shallow ($L = T = 4$) and deep ($L = T = 32/64$) structure. The representations of ResGCN converge in narrow, curve-shaped regions for deep structures. This indicates that the representations converge to a narrow subspace, which is consistent with Corollary 1. Applying our framework (bottom row) addresses this issue, resulting in *spread-out* representations with deeper network structure. This suggests that the application of our framework enhances the expressivity.

## 5.2 Performance Comparison on GNN Variants

We evaluate the performance of GNN variants integrated with our framework to determine the optimal neighborhood scopes for the task on the homophilic (Cora, Citeseer, Pubmed) (Sen et al., 2008) and heterophilic (Chameleon, Cornell, Texas, Wisconsin) (Pei et al., 2020) graphs by comparing with the variants.

For homophilic graphs, we perform both full-supervised (Citeseer, Cora) and semi-supervised (Pubmed) node classification in Table 1. We integrate our inference framework with vanilla GCN, ResGCN (Kipf & Welling, 2016) (GCN with residual skip connections), GAT (Veličković et al., 2017), JKNet (Xu et al., 2018), GCNII (Chen et al., 2020), and ACM-GCN+ (Luan et al., 2022) [2]. Additional baselines we use are GPR-GNN (Chien et al., 2021), and a bayesian GCN variant BBGDC (Hasanzadeh et al., 2020). The implementation details are provided in appendix G. Table 1 shows that the GNN variants with our framework achieve the best performance on four graph datasets and the second best on the remaining three datasets. The results suggest that by jointly inferring neighborhood scope and learning GNN parameters, we boost the overall performance of the GNN models.

---

[2]Integration details are provided in Appendix F

Table 2: Uncertainty calibration comparison between baseline GNN models, an ensemble of the baseline models, and applying our framework in the baseline models. The reported metric is the expected calibration error (ECE ↓). The best result is bolded.

| Baselines | Cora | Citeseer | Pubmed | Chameleon | Cornell | Texas | Wisconsin |
|---|---|---|---|---|---|---|---|
| GCN | 0.14±0.03 | 0.27±0.03 | 0.17±0.02 | 0.11±0.03 | 0.46±0.06 | 0.49±0.09 | 0.30±0.13 |
| GCN Ensemble | **0.02 ± 0.01** | 0.21±0.02 | **0.04±0.01** | 0.04±0.01 | 0.51±0.02 | 0.57±0.02 | 0.25±0.03 |
| Ours+GCN | 0.04±0.02 | **0.12±0.05** | 0.08±0.03 | 0.05±0.01 | 0.48±0.05 | 0.15±0.03 | 0.13±0.05 |
| GCNII | 0.32±0.01 | 0.39±0.02 | **0.04±0.01** | 0.06±0.01 | 0.36±0.03 | 0.13±0.03 | 0.18±0.05 |
| GCNII Ensemble | 0.32±0.01 | 0.39±0.01 | **0.04±0.01** | 0.03±0.01 | 0.34±0.01 | 0.20±0.01 | 0.21±0.02 |
| ACM-GCN+ | 0.19±0.04 | 0.27±0.03 | 0.15±0.03 | 0.04±0.01 | 0.13±0.03 | 0.09±0.02 | 0.09±0.02 |
| ACM-GCN+ Ensemble | 0.17±0.01 | 0.28±0.01 | 0.16±0.01 | 0.05±0.01 | 0.12±0.02 | 0.11±0.03 | 0.12±0.02 |
| Ours+ACM-GCN+ | 0.09±0.03 | 0.13±0.02 | 0.14±0.01 | 0.04±0.01 | **0.10±0.05** | **0.07±0.01** | **0.06±0.01** |

## 5.3 Expressivity with Deep GNN Structures

We evaluate the effects of our inference framework on the expressivity of GNN variants with increasingly deep structures. We apply dropout regularization to the GNN variants in this analysis. The performance over varying numbers of GNN layers is shown in Figure 4. The results show that the overall performance of ResGCN, GAT, and ACM-GCN+ across the datasets suffer a decline when the network depth $L$ becomes large. However, combining our framework with these GNN models to adapt the neighborhood scopes for the node feature learning, we mitigate the problem as indicated by the solid flat curves, showing the robustness of the performance over the increasing truncation level $T$.

In Figure 5, we assess expressivity by visualizing the node representations learned by ResGCN with and without our framework. The t-SNE embeddings of the representations obtained from the last layer of a shallow network ($L = 4$) and deep network ($L = 32/64$) are shown for the Cora and Wisconsin datasets.[3] The representations generated by ResGCN with shallow networks are well-separated into clusters and are spread-out in the latent feature space. However, For deep ResGCN, the cluster separation becomes less distinct, and the representations collapse into a curved-shaped region. This is consistent with Corollary 1. The application of our framework (bottom row) results in comparatively spread-out representations for shallow structure ($T/L = 4$), which is in accordance with Theorem 2. However, the representations remain spread-out even for deep structures, indicating improved expressivity as suggested by Corollary 2. This can be attributed to the property of our framework that decouples the neighborhood scope from the truncation level (i.e., a pre-specified network depth) and allows the ResGCN network depth to adapt as it learns node representations.

## 5.4 Uncertainty Quantification

We assess the uncertainty estimates of GNN variants, their combinations with our framework, and GNN deep assembles. The baselines include the vanilla GCN and the two best performers from Table 1, namely GCNII and ACM-GCN+. The ensemble of the baseline models consists of 10 models trained with different initializations. The metric for assessing uncertainty is expected calibration error (ECE) (Guo et al., 2017).[4] Table 2 shows that compared to the baseline GNN models, integrating our framework improves their uncertainty quantification on both homophilic and heterophilic datasets in most cases. By quantifying the uncertainty of adaptive neighborhood scopes in training via Bayesian inference, our approach enhances uncertainty calibration in four cases and delivers comparable results in the remaining three, compared to the GNN deep ensemble.

## 5.5 Ablation Study

We analyze the contribution of different modeling components of our framework on GCNs. ResGCN is the GCN with residual connections between successive layers. The results in Table 3 show that residual connections is an effective technique, and with dropout regularization (*do*) for feature sampling ResGCN improve

---

[3]More detailed expressivity assessments along with overfitting analysis are in Appendix K and H respectively.

[4]Analysis using $PAvsPU$ metric is in Appendix I.

Table 3: Ablation study of our online inference framework for neighborhood scope adaptation.

| Dataset | Cora | Citeseer | Pubmed |
|---------|------|----------|--------|
| GCN | 85.00±0.10 | 77.23±0.17 | 77.00±0.50 |
| ResGCN | 86.16±0.24 | 77.26±0.10 | 76.66±0.33 |
| ResGCN+*do* | 86.15±0.15 | **78.15±0.30** | 77.66±0.84 |
| Ours+GCN | **86.83±0.13** | 77.90±0.37 | **78.20±0.29** |

the GCN's performance and achieve the best on Citeseer. Furthermore, by adapting the neighborhood scope with beta process, our framework achieves the overall best performance.

Table 4: Performance of baselines and our method on large graph datasets. The metric reported are percentage accuracy for Flickr & ogb-arxiv, and AU-ROC for ogb-proteins. The best result is highlighted in bold and the second-best is underlined. The last column shows the GPU memory usage for one epoch (in Gigabytes) while training the baselines and our method ($S = 3$) on the ogb-protein dataset.

| Dataset | Flickr | ogb-arxiv | ogb-proteins | Memory (ogb-proteins) |
|---------|--------|-----------|--------------|------------------------|
| GCN | 51.44±0.13 | 71.74±0.29 | 0.7251±0.0025 | 9.19 |
| ResGCN | 51.38±0.11 | 72.86±0.16 | 0.7343±0.0016 | 9.88 |
| Ours + ResGCN | 51.73±0.21 | 72.79±0.30 | **0.7572±0.0041** | 10.06 |
| JKNet | **52.56±0.12** | 72.19±0.21 | 0.6966±0.0052 | 10.14 |
| Ours + JKNet | 52.24±0.28 | 72.88±0.09 | 0.7330±0.0068 | 10.41 |
| GCNII | 51.53±0.16 | 72.74±0.16 | 0.7414±0.0070 | 9.91 |
| Ours + GCNII | 51.48±0.14 | **73.06±0.40** | 0.7513±0.0054 | 10.07 |

### 5.6 Performance on Large Datasets

We evaluate baselines and our method on three large datasets: Flickr, ogb-arxiv on multi-class classification, and ogb-proteins on binary classification. The reported metrics are percentage accuracy for multi-class classification and AU-ROC for binary classification settings. Table 4 demonstrates that applying our framework to the baselines results in comparable or significantly better performance, showing that the performance of our framework scales effectively to large datasets.

### 5.7 Training
### Time and Space Complexity Evaluation

For a constant maximum layer width, the time complexity of training a GNN model with depth $L$ is $\mathcal{O}_t = \mathbb{O}(L|\mathcal{V}|^2 + |\mathcal{V}|)$. Let $S$ denote the number of Monte Carlo samples, our method is linearly scalable as $S\mathcal{O}_t$. The space complexity of training the GNN model is $\mathbb{O}(|\mathcal{V}|^2 + L|\mathcal{V}|)$. For our method, the space complexity is $\mathbb{O}(|\mathcal{V}|^2 + SL|\mathcal{V}|)$.

We report the training times of the GNN variants combining with our framework in Table 5. For the inference over neighborhood scopes, the number of samples of **Z** is set to $S = 5$. The results align with

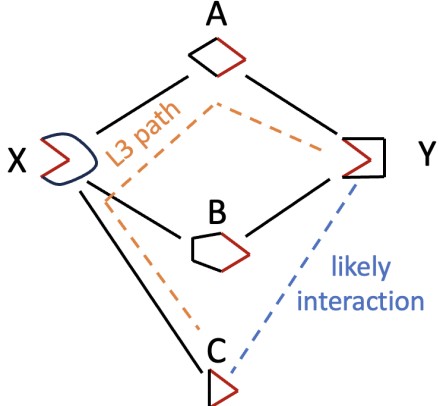

Figure 6: A simplified illustration of a Protein-Protein Interaction (PPI) network. Kovács et al. (2019) provide evidence that proteins that are connected by multiple L3 paths (i.e. 3-hope apart)(C & Y in the diagram) are more likely to interact than others, highlighting the importance of aggregating messages from three-hop neighbors when learning node representations.

our complexity analysis, showing that the training time of our method scales linearly with the number of samples. The convergence of our method with respect to the varying number of samples $S$ is discussed in Appendix L, which also provides an overview of the trade-off between the increased complexity of our method (brought by a higher value of $S$) and the performance. Compared to the baselines, although our method takes extra time for the joint inference in training to infer the best settings of neighborhood scopes, this is justified by the overall performance improvement over the baselines, improved expressivity of baseline GNN, improved uncertainty calibration, and mitigation of overfitting in the baselines (Appendix H).

In the last column of Table 4, we report the GPU memory usage when training the baseline models with and without our framework on the ogb-proteins dataset. The results demonstrate that integrating our framework introduces no significant increase in memory usage. This aligns with our complexity analysis, which shows that for larger graphs (i.e., large $|\mathcal{V}|$), the first term in the space complexity $\mathbb{O}(|\mathcal{V}|^2 + SL|\mathcal{V}|)$ dominates, while the second term contributes minimally to the overall memory load. Since the first term is the same for both the baselines and our approach, memory consumption remains effectively unchanged.

Table 5: Training times (in seconds) of GNN variants with and without our framework for 100 epochs. The number of samples for our method is set to $S = 5$.

| Dataset | Cora | Citeseer | Flickr | ogb-arxiv |
|---|---|---|---|---|
| ResGCN | 0.35 | 0.37 | 2.30 | 5.60 |
| Ours + GCN | 1.14 | 1.30 | 10.12 | 29.20 |
| GCNII | 0.60 | 0.62 | 2.84 | 6.65 |
| Ours + GCNII | 1.21 | 1.18 | 11.64 | 31.60 |

### 5.8 Neighborhood Scope Adaptation on Biomolecular Network

As a case study, we evaluate our model on a real-world biomolecular graph dataset. Specifically, we apply it to the link prediction task in the Protein-Protein Interaction (PPI) dataset. In this graph, nodes represent protein molecules, and edges denote interactions between them. Kovács et al. (2019) provide evidence that proteins that are three hops apart (i.e. that are connected by L3 paths) are likely to interact as shown in Figure 6.

This suggests that, in PPI graphs, aggregating messages from 3-hop neighbors is essential for effective representation learning. To validate the neighborhood scope inference capability of our method in this domain-specific setting, we assess whether it aligns with the findings of Kovács et al. (2019). Specifically, we evaluate whether our framework can infer a neighborhood scope that enables message passing between nodes 3 hops apart.

Given an interaction graph with a fraction of missing positive interactions, the goal is to predict these missing links. For experimental evaluation, we randomly split the interactions into training (70%), validation (10%), and testing (20%) sets. Since the dataset contains only positive interactions, we generate negative samples by randomly selecting node pairs that lack an observed edge, assuming these represent non-interacting proteins.

We train a (GNN) to learn node representations and an interaction between the nodes are presented based on the similarity of representations. The AU-ROC and AUPRC scores for ResGCN and Ours + ResGCN on the link prediction task are (0.907 ±

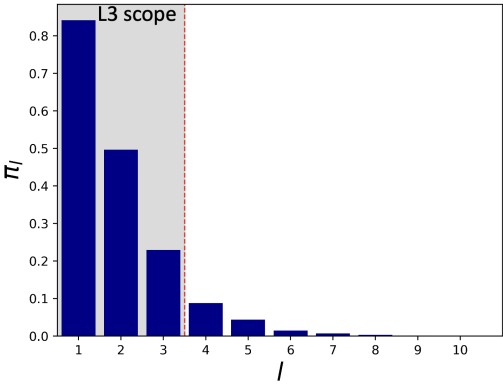

Figure 7: Contribution probabilities learned by our method for the PPI network. The shaded region (L3 scope) corresponds to the neighborhood scope encompassing L3 neighbors. The activation probabilities drop below 0.1 from the fourth hop, showing that the effective neighborhood scope in 3 hops. The neighborhood scope inferred from our method aligns with the findings in Kovács et al. (2019), which suggests that proteins lying 3-hops away are likely to interact.

$0.006, 0.894 \pm 0.002)$ and $(\mathbf{0.918 \pm 0.002}, \mathbf{0.907 \pm 0.003})$, respectively. The learned contribution probabilities are visualized in Figure 7. The contribution probabilities drop below 0.1 starting from the fourth hop, indicating that effective message aggregation occurs from nodes within the 3-hop range. This suggests that the inferred neighborhood scope allows the interaction with the neighbors 3-hops away, aligning with the findings in Kovács et al. (2019).

## 6 Discussion and Conclusion

We propose a general automatic neighborhood scope adaption method compatible with various GNN models and boosting the overall performance and uncertainty estimation. We show that the application of our framework improves the expressivity with deep network structures. Furthermore, we demonstrated the neighborhood scope inference capability of our method on a real-world PPI network. Our future work entails adopting our neighborhood adaptation strategy for more complex GNN architectures, such as graph transformer networks (Yun et al., 2019). Another future direction is relaxing the finite truncation constraint in the variational distributions by incorporating the Russian roulette method (Xu et al., 2019).

### Broader Impact Statement

This work advances Graph Neural Networks, a machine learning model. All datasets used in the paper are publicly available and the experiments do not involve human subjects. The work has no specific societal impact beyond the ones that come with developing a machine learning model.

### Acknowledgement

This work is supported by the National Science Foundation under NSF Award No. 2045804 and the National Institute of General Medical Sciences of the National Institutes of Health under Award No. R35GM156653. We acknowledge Research Computing at the Rochester Institute of Technology (RC) for providing computational resources.

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

## A   Appendix

## B   Proof of Lemma 1

Restating the notations and assumptions in the main text, for ease of notation we use $N$ to denote the number of nodes in the graph ($N = |\mathcal{V}|$). For a symmetric adjacency matrix $\mathbf{A}$, the eigenvectors are perpendicular. Let $\lambda_1, \ldots, \lambda_N$ are the eigenvalues of $\mathbf{A}$ sorted in ascending order, and let the multiplicity of largest eigenvalue $\lambda_N$ is M i.e $\lambda_1 < \ldots, \lambda_{N-M} < \lambda_{N-M+1} = \cdots = \lambda_N$. We also assume that the adjacency matrix is normalized and possesses positive eigenvalues, with the maximum eigenvalue capped at 1.

Let, $\{e_m\}_{m=N-M+1,\ldots,N}$ be the orthonormal basis of the subspace $U$ corresponding the the eigenvalues $\{\lambda_m\}_{m=N-M+1,\ldots,N} = 1$, and $\{e_m\}_{m=1,\ldots,N-M}$ be the orthonormal basis for $U^\perp$. Consequently, $\mathbf{H} \in \mathbb{R}^{N \times O}$ can be expressed as $\mathbf{H} = \sum_{m=1}^N e_m \otimes w_m$ where $w_m \in \mathbb{R}^O$.

The difference between a normal multi-layer perception (MLP) layer and GCN layer lies in the multiplication of the layer output with adjacency matrix $\mathbf{A}$. And this repeated multiplication with $\mathbf{A}$ at every layer is the cause of the collapse of node features into a subspace (Oono & Suzuki, 2019). To analyze how this repeated multiplication affects a GCN with residual connections, in our analysis we simplify a GCN layer to a multiplication of features $\mathbf{H}$ with the adjacency matrix $\mathbf{A}$.

If $\theta$ is the angle made by $\mathbf{H}_L$ with subspace $U$,

$$\tan \theta = \frac{d_{\mathcal{M}}(\mathbf{H}_L)}{|\mathbf{P}|} \tag{13}$$

$$\theta = \tan^{-1}\left[\frac{d_{\mathcal{M}}(\mathbf{H}_L)}{|\mathbf{P}|}\right] \tag{14}$$

For ResGCN, a layer is defined as:

$$\mathbf{H}_L^{(Res)} = f(\mathbf{H}_{L-1}^{(Res)}) + \mathbf{H}_{L-1}^{(Res)}$$

Representing a layer by its adjacency matrix,

$$\mathbf{H}_L^{(Res)} = \mathbf{A}\mathbf{H}_{L-1}^{(Res)} + \mathbf{H}_{L-1}^{(Res)}$$

$$\mathbf{H}_L^{(Res)} = (\mathbf{A} + \mathbf{I})\mathbf{H}_{L-1}^{(Res)}$$

Solving this recurrence, we get:

$$\mathbf{H}_L^{(Res)} = (\mathbf{A} + \mathbf{I})^L \mathbf{H}_0^{(Res)} \tag{15}$$

Expanding Eqn. (15) in terms of $\{w_m\}$, we get:

$$\mathbf{H}_L^{(Res)} = \sum_{m=1}^N e_m \otimes (\lambda_m + 1)^L w_m$$

$$\tag{16}$$

The distance from the subspace $U$ is:

$$
\begin{aligned}
d_{\mathcal{M}}^2(\mathbf{H}_L^{(Res)}) &= \sum_{m=1}^{N-M} ||(\lambda_m + 1)^L w_m||^2 \\
&= \sum_{m=1}^{N-M} ||(1 + \frac{1}{\lambda_m})^L (\lambda_m)^L w_m||^2 \\
&= \sum_{m=1}^{N-M} (1 + \frac{1}{\lambda_m})^{2L} ||(\lambda_m)^L w_m||^2
\end{aligned}
\tag{17}
$$

Since, $0 < \lambda_m < 1, (1 + \frac{1}{\lambda_m}) > 2.$ Then,

$$
\begin{aligned}
\implies d_{\mathcal{M}}^2(\mathbf{H}_L^{(Res)}) &\geq 2^{2L} d_{\mathcal{M}}^2(\mathbf{H}_L) \\
\implies d_{\mathcal{M}}(\mathbf{H}_L^{(Res)}) &\geq 2^L d_{\mathcal{M}}(\mathbf{H}_L)
\end{aligned}
\tag{18}
$$

Similarly,

$$
\begin{aligned}
|\mathbf{P}_L^{(Res)}|^2 &= \sum_{m=N-M+1}^{N} ||(\lambda_m + 1)^L w_m||^2 \\
&\text{Since, } \lambda_m = 1 \text{ for } N - M + 1 \leq m \leq N, \\
|\mathbf{P}_L^{(Res)}|^2 &= \sum_{m=N-M+1}^{N} ||2^L w_m||^2 \\
&= 2^{2L} \sum_{m=N-M+1}^{N} ||w_m||^2 \\
&= 2^{2L} |\mathbf{P}_L|^2 \\
\implies |\mathbf{P}_L^{(Res)}| &= 2^L |\mathbf{P}_L|
\end{aligned}
\tag{19}
$$

The angular region spanned is:

$$
\begin{aligned}
\tan \theta_L^{(Res)} &= \frac{d_{\mathcal{M}}(\mathbf{H}_L^{(Res)})}{|\mathbf{P}_L^{(Res)}|} \\
\implies \tan \theta_L^{(Res)} &\geq \frac{2^L d_{\mathcal{M}}(\mathbf{H}_L)}{2^L |\mathbf{P}_L|} \\
\implies \tan \theta_L^{(Res)} &\geq \tan \theta_L \\
\implies \theta_L^{(Res)} &\geq \theta_L
\end{aligned}
\tag{20}
$$

### B.1 Corollary 1

From equation (5) :

$$
\begin{aligned}
d_{\mathcal{M}}^2(\mathbf{H}_L^{(Res)}) &= \sum_{m=1}^{N-M} (1 + \frac{1}{\lambda_m})^{2L} ||(\lambda_m)^L w_m||^2 \\
&= \sum_{m=1}^{N-M} (1 + \frac{1}{\lambda_m})^{2L} (\lambda_m)^2 ||(\lambda_m)^{L-1} w_m||^2 \\
&= \sum_{m=1}^{N-M} (\lambda_m)^2 (1 + \frac{1}{\lambda_m})^2 (1 + \frac{1}{\lambda_m})^{2(L-1)} ||(\lambda_m)^{L-1} w_m||^2 \\
&= \sum_{m=1}^{N-M} (\lambda_m + 1)^2 (1 + \frac{1}{\lambda_m})^{2(L-1)} ||(\lambda_m)^{L-1} w_m||^2 \\
&\leq \sum_{m=1}^{N-M} 2^2 (1 + \frac{1}{\lambda_m})^{2(L-1)} ||(\lambda_m)^{L-1} w_m||^2; \quad [\text{Since, } \lambda_m \leq 1] \\
&= 2^2 \ d_{\mathcal{M}}^2(\mathbf{H}_{L-1}^{(Res)}) \\
\implies d_{\mathcal{M}}(\mathbf{H}_L^{(Res)}) &\leq 2 \ d_{\mathcal{M}}(\mathbf{H}_{L-1}^{(Res)})
\end{aligned}
\tag{21}
$$

The angular region spanned for $L$ layers is:

$$
\begin{aligned}
\tan \theta_L^{(Res)} &= \frac{d_{\mathcal{M}}(\mathbf{H}_L^{(Res)})}{|\mathbf{P}_L^{(Res)}|} \\
\implies \tan \theta_L^{(Res)} &\leq \frac{2 \ d_{\mathcal{M}}(\mathbf{H}_{L-1}^{(Res)})}{2^L |\mathbf{P}_L|} \\
\implies \tan \theta_L^{(Res)} &\leq \frac{d_{\mathcal{M}}(\mathbf{H}_{L-1}^{(Res)})}{2^{L-1} |\mathbf{P}_{L-1}|} \quad [\text{Since, } |\mathbf{P}_{L-1}| = |\mathbf{P}_L|] \\
\implies \tan \theta_L^{(Res)} &\leq \tan \theta_{L-1}^{(Res)} \\
\implies \theta_L^{(Res)} &\leq \theta_{L-1}^{(Res)}
\end{aligned}
\tag{22}
$$

## C  Proof of Theorem 2

First, we prove a simple relation. For $0 \leq \pi, \lambda \leq 1$

$$
\begin{aligned}
&\frac{\lambda \pi + 1}{\lambda + 1} - \frac{\pi + 1}{2} \\
&= \frac{2(\lambda \pi + 1) - (\lambda + 1)(\pi + 1)}{2(\lambda + 1)} \\
&= \frac{(1 - \lambda)(1 - \pi)}{2(\lambda + 1)} \\
&\geq 0 \\
&\implies \frac{\lambda \pi + 1}{\lambda + 1} \geq \frac{\pi + 1}{2}
\end{aligned}
\tag{23}
$$

A layer in BNA-GCN is represented as:

$$
\mathbf{H}_L^{(BNA)} = f(\mathbf{H}_{L-1}^{(BNA)}) \otimes \mathbf{z}_L + \mathbf{H}_{L-1}^{(BNA)}
\tag{24}
$$

Since $\mathbf{Z}$ is a random variable, we calculate the expectation as:

$$\begin{aligned}
\mathbb{E}[\mathbf{H}_L^{(BNA)}] &= f(\mathbf{H}_{L-1}^{(BNA)}) \otimes \mathbb{E}[\mathbf{z}_L] + \mathbf{H}_{L-1}^{(BNA)} \\
&= f(\mathbf{H}_{L-1}^{(BNA)}) \otimes \pi_L + \mathbf{H}_{L-1}^{(BNA)}
\end{aligned}$$

Representing layer by the adjacency matrix, (dropping the expectation notation for convenience),

$$\begin{aligned}
\mathbf{H}_L^{(BNA)} &= \mathbf{A}\mathbf{H}_{L-1}^{(BNA)} \otimes \pi_L + \mathbf{H}_{L-1}^{(BNA)} \\
&= (\mathbf{A}\pi_L + 1)\mathbf{H}_{L-1}^{(BNA)}
\end{aligned}$$

Solving this recurrence, we get:

$$\mathbf{H}_L^{(BNA)} = \prod_{l=1}^{L}(\mathbf{A}\pi_l + 1)\mathbf{H}_0^{(BNA)}$$

Expanding in terms of $w_m$, we get:

$$\mathbf{H}_L^{(BNA)} = \sum_{m=1}^{N-M}\prod_{l=1}^{L}(\lambda_m\pi_l + 1)w_m$$

The distance from subspace $U$ is:

$$\begin{aligned}
d_{\mathcal{M}}^2(\mathbf{H}_L^{(BNA)}) &= \sum_{m=1}^{N-M} ||\prod_{l=1}^{L}(\lambda_m\pi_l + 1)w_m||^2 \\
&= \sum_{m=1}^{N-M}\left[\prod_{l=1}^{L}(\lambda_m\pi_l + 1)\right]^2 ||w_m||^2 \\
&= \sum_{m=1}^{N-M}\left[\prod_{l=1}^{L}\frac{\lambda_m\pi_l + 1}{\lambda_m + 1}\right]^2 ||(\lambda_m + 1)^L w_m||^2 \\
&\geq \left[\prod_{l=1}^{L}\frac{\pi_l + 1}{2}\right]^2 \sum_{m=1}^{N-M} ||(\lambda_m + 1)^L w_m||^2 \quad \text{[From Eqn. 23]} \\
&\geq \frac{\prod_{l=1}^{L}(\pi_l + 1)^2}{2^{2L}}d_{\mathcal{M}}^2(\mathbf{H}_L^{(Res)}) \\
\implies d_{\mathcal{M}}(\mathbf{H}_L^{(BNA)}) &\geq \frac{\prod_{l=1}^{L}(\pi_l + 1)}{2^L}d_{\mathcal{M}}(\mathbf{H}_L^{(Res)})
\end{aligned} \tag{25}$$

Similarly,

$$|\mathbf{P}_L^{(BNA)}|^2 = \sum_{m=N-M+1}^{N} ||\prod_{l=1}^{L}(\lambda_m \pi_l + 1)w_m||^2$$

$$\text{Since, } \lambda_m = 1 \text{ for } N - M + 1 \le m \le N,$$

$$= \sum_{m=N-M+1}^{N} ||\prod_{l=1}^{L}(\pi_l + 1)w_m||^2$$

$$= \prod_{l=1}^{L}(\pi_l + 1)^2 \sum_{m=N-M+1}^{N} ||w_m||^2$$

$$= \frac{\prod_{l=1}^{L}(\pi_l + 1)^2}{2^{2L}} \sum_{m=N-M+1}^{N} 2^{2L}||w_m||^2$$

$$= \frac{\prod_{l=1}^{L}(\pi_l + 1)^2}{2^{2L}} |\mathbf{P}_L^{Res}|^2$$

$$\implies |\mathbf{P}_L^{(BNA)}| = \frac{\prod_{l=1}^{L}(\pi_l + 1)}{2^L} |\mathbf{P}_L^{Res}| \tag{26}$$

The angular region spanned is:

$$\tan\theta_L^{(BNA)} = \frac{d_{\mathcal{M}}(\mathbf{H}_L^{(BNA)})}{|\mathbf{P}_L^{(BNA)}|}$$

$$\text{Substituting the values from Eqn. 25 and 26 and simplifying}$$

$$\implies \tan\theta_L^{(BNA)} \ge \frac{d_{\mathcal{M}}(\mathbf{H}_L^{(Res)})}{|\mathbf{P}_L^{(Res)}|}$$

$$\implies \tan\theta_L^{(BNA)} \ge \tan\theta_L^{(Res)}$$

$$\implies \theta_L^{(BNA)} \ge \theta_L^{(Res)} \tag{27}$$

## C.1 Corollary 2

By the definition of $l^{ns}$, $\mathbf{z}_l = \mathbf{0}$ for $l > l^{ns}$. From Eqn. (24), we have:

$$\mathbf{H}_L^{(BNA)} = f(\mathbf{H}_{L-1}^{(BNA)}) \otimes \mathbf{z}_L + \mathbf{H}_{L-1}^{(BNA)}$$

$$\text{For } l = l^{ns} + 1,$$

$$\mathbf{H}_{l^{ns}+1}^{(BNA)} = f(\mathbf{H}_{l^{ns}}^{(BNA)}) \otimes \mathbf{z}_{l^{ns}+1} + \mathbf{H}_{l^{ns}}^{(BNA)}$$

$$\mathbf{H}_{l^{ns}+1}^{(BNA)} = \mathbf{H}_{l^{ns}}^{(BNA)}$$

$$\text{Generalizing the relation:}$$

$$\mathbf{H}_l^{(BNA)} = \mathbf{H}_{l^{ns}}^{(BNA)} \quad \text{for } l > l^{ns}$$

$$\text{Therefore,}$$

$$\theta_l^{(BNA)} = \theta_{l^{ns}}^{(BNA)} \quad \text{for } l > l^{ns}$$

## D    Algorithmic Description

The algorithm of our proposed framework is in Algorithm 1.

---

**Algorithm 1** Training of our proposed method

---

**Input** Graph $\mathcal{G}, D, S$, prior parameters $\alpha, \beta$.
**Initialize** Variational parameters $\{a_t, b_t\}_{t=1}^T$

1: Draw $S$ samples of network structures $\{\mathbf{Z}_s\}_{s=1}^S$ from $q(\mathbf{Z}, \boldsymbol{\nu})$
2: **for** s = 1, \ldots, S **do**
3:     Compute the neighborhood scope $l^{ns}$ from $\mathbf{Z}_s$ (see section 3.3).
4:     Compute $\log p(D|\mathbf{Z}_s, \mathbf{W}, \mathcal{G})$ with $l^{ns}$ GNN layers using equation 5.
5: **end for**
6: Compute ELBO using equation 8 .
7: Update $\{a_t, b_t\}_{t=1}^{l^{ns}}$ and $\{\mathbf{W}\}_{t=1}^{l^{ns}}$ using backpropagation.

---

## E    Structural Diagram of the Proposed Framework

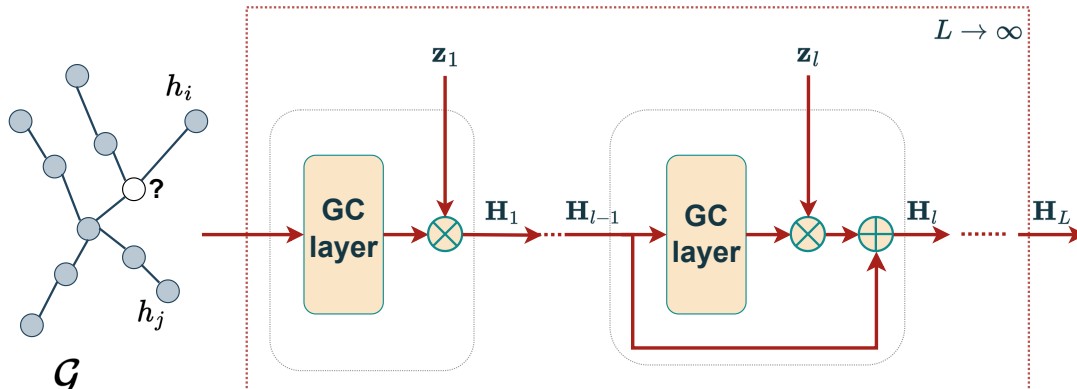

Figure 8: The block diagram of our proposed model with a potentially infinite number of hidden layers in the GCN corresponds to a potentially infinite scope for message aggregation. In practice, a sufficiently large number of hidden layers is set. The input to the network is a graph $\mathcal{G}$ with edges $\mathcal{E}$ between entities $\mathcal{V}$ and feature matrix $\mathbf{H}$. The gray-colored circles are the nodes with known labels and blank circles are the nodes with unknown labels. The feature output from a GC layer is sampled using the binary vector $\mathbf{z}_l$. The model also has a residual skip connection between the layers.

## F    Integrating the Framework With the GCN Variants

To integrate our inference framework in vanilla GCN, we multiply the layer output with binary vector $\mathbf{z}_l$ and add a skip connection between the layers:

$$\mathbf{H}_l = \sigma(\widehat{\mathbf{A}}\mathbf{H}_{l-1}\mathbf{W}_l) \bigotimes \mathbf{z}_l + \mathbf{H}_{l-1}, \quad l \in \{1, 2, \ldots \infty\} \tag{28}$$

The difference between GCN and GAT lies in the calculation of the attention coefficient for message aggregation. However, since they share similar network structure, integrating our framework in GAT is straightforward and is similar to GCN. Similarly JKNet and GCN also share similar network structure and hence our framework is integrated in the same way. The aggregation layer in JKNet is kept unchanged while integrating our framework.

GCNII had two additional components in the network, the initial residual connection and identity mapping. A GCNII layer is defined as:

$$\mathbf{H}_l = \sigma\Big(\big((1-\alpha)\widehat{\mathbf{A}}\mathbf{H}_{l-1} + \alpha\mathbf{H}_1\big)\big((1-\beta_l)\mathbf{I} + \beta_l\mathbf{W}_l\big)\Big) \tag{29}$$
$$\text{where,} \quad \beta_l = log(\lambda/l + 1)$$

We have a couple of options to integrate our framework with GCNII. The first is just incorporating an initial residual connection as follows (used for homophilic datasets):

$$\mathbf{H}_l = \sigma(((1-\alpha_t)\widehat{\mathbf{A}}\mathbf{H}_{l-1} + \alpha_t\mathbf{H}_1)\mathbf{W}_l)\bigotimes \mathbf{z}_l + \mathbf{H}_{l-1} \tag{30}$$

Secondly, we can also incorporate the identity mapping module as follows (used for heterophilic datasets):

$$\mathbf{H}_l = \sigma\Big(\big((1-\alpha_t)\widehat{\mathbf{A}}\mathbf{H}_{l-1} + \alpha_l\mathbf{H}_1\big)\big((1-\beta_l)\mathbf{I} + \beta_l\mathbf{W}_l\big)\Big)\bigotimes \mathbf{z}_l \tag{31}$$

where $\alpha_t$ is the teleport probability similar to GCNII, a subsrcript $t$ is introduced to differentiate if from the prior parameter $\alpha$.

## G   Implementation Details

### G.1   Datasets

We use the publicly available datasets for experimentation which includes the three homophilic citation graphs: Citeseer, Cora & Pubmed and four heterophilic graphs: Chameleon, Cornell, Texas, and Wisconsin. The dataset details are in Table 6. The experiments are carried out on NVIDIA A100-PCIE-40GB and NVIDIA RTX A5000 GPUs.

Table 6: Dataset details

| Dataset | #Nodes | #Edges | #Classes | #Features |
|---|---|---|---|---|
| **Cora** | 2708 | 5429 | 7 | 1433 |
| **Citeseer** | 3327 | 4732 | 6 | 3703 |
| **Pubmed** | 19717 | 44338 | 3 | 500 |
| **Chameleon** | 2277 | 36101 | 4 | 2325 |
| **Cornell** | 183 | 295 | 5 | 1703 |
| **Texas** | 183 | 309 | 5 | 1703 |
| **Wisconsin** | 251 | 499 | 5 | 1703 |
| **Flickr** | 89250 | 899756 | 7 | 500 |
| **ogb-arxiv** | 169343 | 1166243 | 40 | 128 |
| **ogb-proteins** | 132534 | 39561252 | 112 | 8 |

### G.2   Hyperparameter Details for Table 1

#### G.2.1   Homophilic Graphs (Citeseer, Cora, Pubmed)

We used the standard fixed split for the homophilic graphs as introduced in (Yang et al., 2016). The general setup for the experiments (unless mentioned otherwise) including the width of hidden layers ($O$), learning rate (lr), and activation function (act) are detailed in Table 7. The value of dropout and learning rate is set as suggested in (Kipf & Welling, 2016). The hyperparameter search for the layers is done in the range [2, 4, 6, 8, 10]. We applied dropedge for the homophilic graphs on the baselines and our method and tuned the dropedge rate over [5%, 10%, 20%, 30%]. For JKNet, $MaxPool$ is used as an

aggregator function. In the case of GAT, we faced an out-of-memory (OOM) error during model training when using the complete graph in a single batch. To address this issue, we employed the ShaDowKHopSampler (Dgl) as per (Zeng et al., 2021), enabling mini-batch training. Each mini-batch was configured with a batch size of 32, and we sampled a maximum of 10 neighbors within a range of two hops.

We report the mean and variance of the accuracy metric over 4 random trials. In addition to the general configuration, Table 8 presents the specific hyperparameter settings. For GCNII and ACM-GCN+, the hyperparameters were configured following the recommendations in the original implementation. In our framework, we fine-tuned the prior parameters $\alpha$ and $\beta$ within the ranges [2, 5, 10, 15] and [2, 4, 6], respectively.

Table 7: General hyperparameter setup for baseline methods and our method for Table 1.

| General hyperparameter setup | |
|---|---|
| $O$ | 128 |
| epochs | 500 |
| patience | 100 |
| lr | 1e-2 |
| dropout | 0.5 |
| act | *ReLU* |
| optimizer | Adam |

Table 8: Implementation details of baselines and our method for the homophilic datasets (*de* and *do* are the dropedge and dropout rates respectively).

| Dataset | Methods | Hyperparameter details |
|---|---|---|
| Cora | GCN | $de = 0.3$ |
| | ResGCN | $de = 0.3$ |
| | GAT | $de = 0.3$ |
| | JKNet | $de = 0.1$ |
| | GCNII | $de = 0.05, \alpha = 0.1, \lambda = 0.5$ |
| | ACM-GCN+ | $de = 0.2, do = 0.7$ |
| | Ours+ResGCN | $de = 0.1, S = 5, \alpha = 5, \beta = 2$ |
| | Ours+GAT | $de = 0.0, S = 5, \alpha = 5, \beta = 2$ |
| | Ours+JKNet | $de = 0.2, S = 5, \alpha = 5, \beta = 2$ |
| | Ours+GCNII | $de = 0.0, S = 5, \alpha = 5, \beta = 2, \alpha_t = 0.1$ |
| | Ours + ACM-GCN+ | $de = 0.2, \alpha = 10, \beta = 2,$ |
| Citeseer | GCN | $de = 0.2$ |
| | ResGCN | $de = 0.2$ |
| | GAT | $de = 0.1$ |
| | JKNet | $de = 0.2$ |
| | GCNII | $de = 0.1, \alpha = 0.1, \lambda = 0.6$ |
| | ACM-GCN+ | $de = 0.2, do = 0.2$ |
| | Ours+ResGCN | $de = 0.2, S = 5, \alpha = 5, \beta = 2$ |
| | Ours+GAT | $de = 0.0, S = 5, \alpha = 5, \beta = 2$ |
| | Ours+JKNet | $de = 0.1, S = 5, \alpha = 5, \beta = 2$ |
| | Ours+GCNII | $de = 0.0, S = 5, \alpha = 2, \beta = 2, \alpha_t = 0.1$ |
| | Ours + ACM-GCN+ | $de = 0.2, \alpha = 10, \beta = 2,$ |
| Pubmed | GCN | $de = 0.3$ |
| | ResGCN | $de = 0.3$ |
| | GAT | $de = 0.05$ |
| | JKNet | $de = 0.05$ |
| | GCNII | $de = 0.1, \alpha = 0.1, \lambda = 0.4$ |
| | ACM-GCN+ | $de = 0.2, do = 0.3$ |
| | Ours+ResGCN | $de = 0.1, S = 5, \alpha = 5, \beta = 2$ |
| | Ours+GAT | $de = 0.0, S = 5, \alpha = 5, \beta = 2$ |
| | Ours+JKNet | $de = 0.1, S = 5, \alpha = 2, \beta = 2$ |
| | Ours+GCNII | $de = 0.0, S = 5, \alpha = 5, \beta = 2, \alpha_t = 0.1$ |
| | Ours + ACM-GCN+ | $de = 0.2, \alpha = 10, \beta = 2,$ |

### G.2.2 Heterophilic Graphs
### (Chameleon, Cornell, Texas, Wisconsin)

For heterophilic datasets, we adopt the 3:1:1 split for the train, validation and test sets respectively as in (Luan et al., 2022). The baselines except GPR-GCN were implemented following the hyperparameter settings in (Luan et al., 2022). For GPR-GCN, the we adopted the results reported in (Luan et al., 2022). The hyperparameters setup when integrating our framework with the baselines is detailed in Table 9.

Table 9: Implementation details of baselines and our method for the heterophilic datasets ($wd$ is the weight decay rate).

| Dataset | Methods | Hyperparameter details |
|---|---|---|
| Chameleon | Ours+ResGCN | $lr = 0.01, wd = 10^{-5}, O = 64, T = 2, S = 5, \alpha = 10, \beta = 2$ |
| | Ours+JKNet | $lr = 0.01, wd = 10^{-5}, O = 64, T = 2, S = 5, \alpha = 10, \beta = 2$ |
| | Ours+GCNII | $lr = 0.01, wd = 5*10^{-6}, O = 64, T = 4, S = 10, \alpha_t = 0.1, \lambda = 0.5, \alpha = 15, \beta = 2$ |
| | Ours+GAT | $lr = 0.01, wd = 10^{-5}, O = 64, T = 2, S = 5, \alpha = 10, \beta = 2$ |
| | Ours+ACM-GCN+ | $lr = 0.004, wd = 10^{-3}, O = 64, T = 1, S = 1, \alpha = 10, \beta = 2$ |
| Cornell | Ours+ResGCN | $lr = 0.1, wd = 5*10^{-3}, O = 64, T = 2, S = 5, \alpha = 5, \beta = 2$ |
| | Ours+JKNet | $lr = 0.1, wd = 10^{-3}, O = 64, T = 2, S = 5, \alpha = 5, \beta = 2$ |
| | Ours+GCNII | $lr = 0.1, wd = 10^{-3}, O = 64, T = 4, S = 10, \alpha_t = 0.5, \lambda = 0.5, \alpha = 10, \beta = 2$ |
| | Ours+GAT | $lr = 0.1, wd = 10^{-3}, O = 64, T = 2, S = 5, \alpha = 10, \beta = 2$ |
| | Ours+ACM-GCN+ | $lr = 0.01, wd = 10^{-3}, O = 64, T = 1, S = 1, \alpha = 5, \beta = 2$ |
| Texas | Ours+ResGCN | $lr = 0.1, wd = 10^{-3}, O = 32, T = 2, S = 5, \alpha = 5, \beta = 2$ |
| | Ours+JKNet | $lr = 0.1, wd = 10^{-3}, O = 32, T = 2, S = 5, \alpha = 5, \beta = 2$ |
| | Ours+GCNII | $lr = 0.1, wd = 10^{-3}, O = 64, T = 4, S = 10, \alpha_t = 0.5, \lambda = 0.5, \alpha = 10, \beta = 2$ |
| | Ours+GAT | $lr = 0.1, wd = 10^{-3}, O = 32, T = 2, S = 5, \alpha = 10, \beta = 2$ |
| | Ours+ACM-GCN+ | $lr = 0.05, wd = 10^{-3}, O = 64, T = 1, S = 1, \alpha = 5, \beta = 2$ |
| Wisconsin | Ours+ResGCN | $lr = 0.1, wd = 10^{-3}, O = 32, T = 2, S = 5, \alpha = 5, \beta = 2$ |
| | Ours+JKNet | $lr = 0.1, wd = 10^{-3}, O = 32, T = 2, S = 5, \alpha = 5, \beta = 2$ |
| | Ours+GCNII | $lr = 0.01, wd = 10^{-3}, O = 64, T = 8, S = 10, \alpha_t = 0.5, \lambda = 0.5, \alpha = 10, \beta = 2$ |
| | Ours+GAT | $lr = 0.1, wd = 10^{-3}, O = 32, T = 2, S = 5, \alpha = 10, \beta = 2$ |
| | Ours+ACM-GCN+ | $lr = 0.05, wd = 10^{-3}, O = 64, T = 1, S = 1, \alpha = 5, \beta = 2$ |

### G.2.3 Hyperparameter Details for Table 4

In Table 4, we evaluate the models on three large graphs: Flickr (Zeng et al., 2020), ogb-arxiv & ogb-proteins (Hu et al., 2020). We follow original train/validation/test split as described in the original papers. The general settings are as described in Table 7. Additional hyperparameter details are provided in Table 10. For the ogb-arxiv dataset, empirically we found that replacing the masking of node features with $\mathbf{Z}_l$ by multiplying the batch-normalized features with the activation probabilities of each layer $\pi_l$ results in better performance.

Table 10: Implementation details for the large graph datasets.

| Dataset | Methods | Hyperparameter details |
|---|---|---|
| Flickr | Ours+ResGCN | $S = 5, \alpha = 5, \beta = 2$ |
| | Ours+JKNet | $S = 5, \alpha = 5, \beta = 2$ |
| | Ours+GCNII | $S = 5, \alpha = 5, \beta = 2, \alpha_t = 0.1$ |
| ogb-arxiv | Ours+ResGCN | $S = 5, \alpha = 25, \beta = 2$ |
| | Ours+JKNet | $S = 3, \alpha = 20, \beta = 2$ |
| | Ours+GCNII | $S = 5, \alpha = 25, \beta = 2, \alpha_t = 0.5$ |
| ogb-proteins | Ours+ResGCN | $S = 3, \alpha = 25, \beta = 2$ |
| | Ours+JKNet | $S = 3, \alpha = 25, \beta = 2$ |
| | Ours+GCNII | $S = 3, \alpha = 25, \beta = 2, \alpha_t = 0.1$ |

## H  Overfitting Analysis

We analyze overfitting in the GCN variants with and without our framework in Figure 9. The results suggest that the variants ResGCN, JKNet, and GAT trained with dropout suffer from overfitting problems as indicated by the increasing value of their validation loss at higher number of epochs. The issue is alleviated by integrating these variants with our framework. GCNII is already robust to the overfitting problem. Application of our framework in GCNII does not have any significant effect on the validation loss.

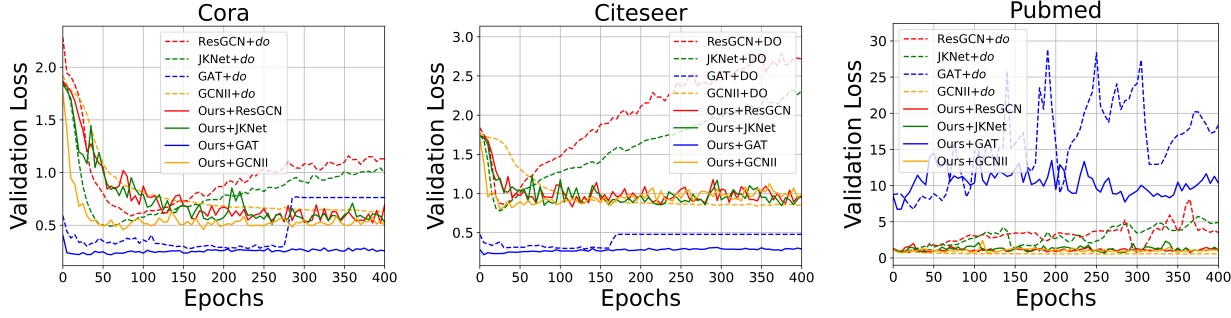

Figure 9: Validation loss for different GCN variants with and without the application of our framework.

## I  Uncertainty Analysis

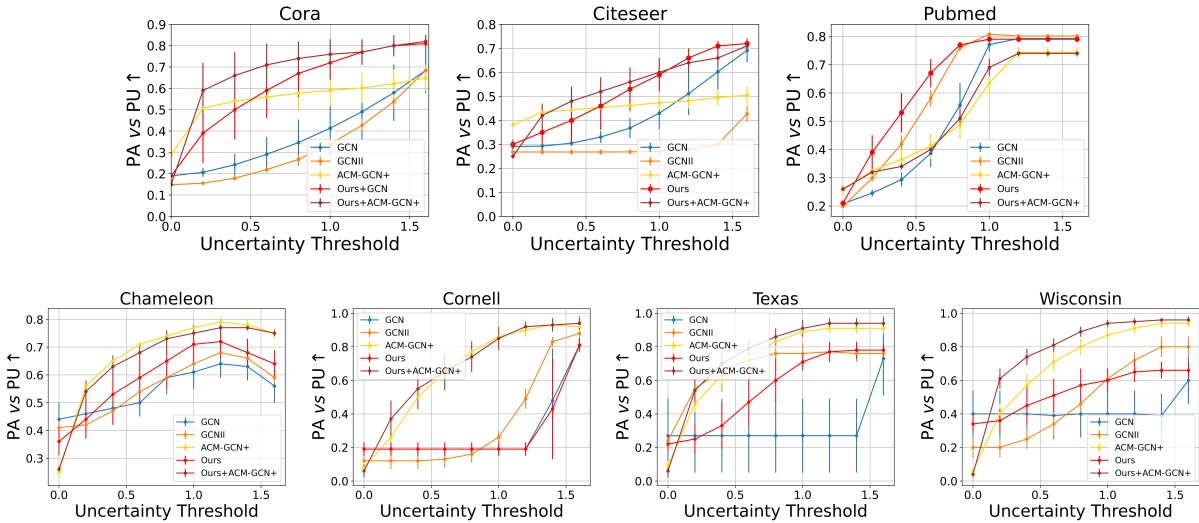

Figure 10: Evaluating the uncertainty estimation of models. The reported metric is $PAvsPU$ (higher values are preferable) plotted against increasing uncertainty thresholds.

In the main text, we evaluated uncertainty calibration of models using the ECE metric. For this study, we performed semi-supervised learning on the homophilic datasets and full supervised learning on the heterophilic datasets. The dataset splits are as defined in sections 6.2.1 and 6.2.2. Here, we first detail the ECE metric and then extend this study by evaluating uncertainty calibration using the $PAvsPU$ metric (Mukhoti & Gal, 2018; Hasanzadeh et al., 2020).

### I.1 Expected Calibration Error (ECE)

Expected Calibration Error (Guo et al., 2017) approximates the difference between predictive confidence and empirical accuracy. First, the predicted confidence $\hat{p}_i$ is partitioned into $I$ equally-spaced bins ($\hat{p}_i = \max \hat{y}_i$, $\hat{y}_i$ is the softmax output). Then ECE is the weighted average of miscalibration in each bin.

$$\text{ECE} = \sum_{i=1}^{I} \frac{|B_i|}{N} |\text{acc}(B_i) - \text{conf}(B_i)| \tag{32}$$

with the number of samples $N$, accuracy of the bin $B_i$

$$\text{acc}(B_i) = \frac{1}{|B_i|} \sum_{i \in B_i} \mathbf{1}[y_i = argmax\{\hat{y}_i\}]$$

and confidence of the bin $B_i$

$$\text{conf}(B_i) = \frac{1}{|B_i|} \sum_{i \in B_i} \hat{p}_i$$

### I.2 Assessing Uncertainty Calibration using the *PAvsPU* metric

To quantify uncertainty, we calculate the entropy of the output softmax distribution. For calculating the metric, we first set an uncertainty threshold. Predictions with uncertainty values below the threshold are classified as *certain* predictions, while those with uncertainty values above the threshold are classified as *uncertain* predictions. The count of *accurate* and *certain* predictions made by the model for a given dataset is denoted as $n_{ac}$. Similarly, the count of *inaccurate* and *uncertain* predictions are denoted as $n_{iu}$. Finally, the metric $PAvsPU$ (Mukhoti & Gal, 2018; Hasanzadeh et al., 2020) is defined as:

$$PAvsPU = (n_{ac} + n_{iu})/(n_{ac} + n_{au} + n_{ic} + n_{iu}) \tag{33}$$

where $n_{au}$ is the count of *accurate* and *uncertain* predictions and $n_{ic}$ the count of *inaccurate* and *certain* predictions. The $PAvsPU$ metric assumes that the model has reliably estimated uncertainty when the predictions are *accurate* and *certain* as well as *inaccurate* and *uncertain*. It measures the proportion of predictions with reliable uncertainty estimation. Higher values of the metric indicates reliable uncertainty estimation.

Figure 10 shows that our method combined with GCN and ACM-GCN+ outperform other baselines in most cases.

## J Concrete Bernoulli Distribution

The probability density function of a Concrete Bernoulli Distribution is:

$$\text{ConBer}(z_{ot}|\pi_t) = \tau \frac{\pi_t(z_{ot})^{-\tau-1}(1-\pi_t)(1-z_{ot})^{-\tau-1}}{(\pi_t(z_{ot})^{-\tau} + (1-\pi_t)(1-z_{ot})^{-\tau})^2} \tag{34}$$

where $\tau$ controls the distribution smoothness. We thus generate the VAE architecture samples $\mathbf{Z}_s$ by first sampling from a logistic distribution, and then putting the samples through a logistic function:

$$z_{ot} = \frac{1}{1 + \exp(-\tau^{-1}(\log \pi_t - \log(1-\pi_t) + \epsilon))} \tag{35}$$
$$\text{where, } \epsilon \sim \text{Logistic}(0,1)$$

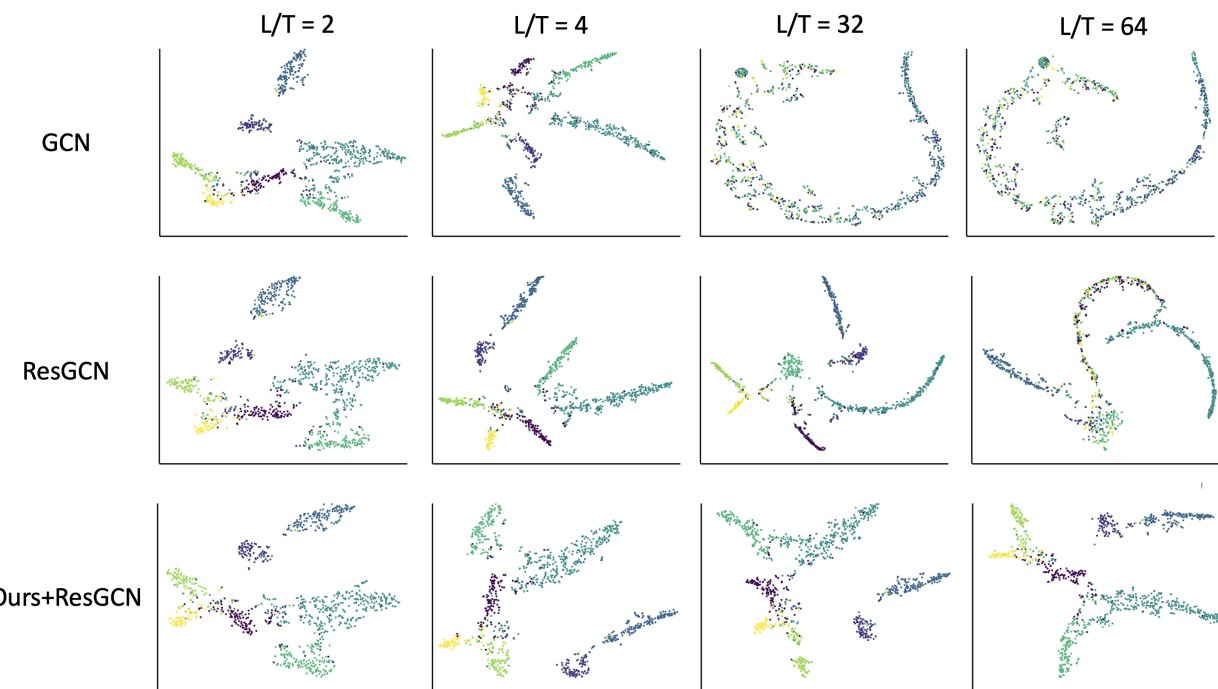

Figure 11: TSNE visualization of the learned node representations by GCN, ResGCN and our framework for shallow ($L = T = \{2, 4\}$) and deep ($L = T = \{32, 64\}$) structures on the Cora dataset. As observed in the top (GCN) and middle (ResGCN) rows, the representations converge in narrow curve-shaped regions for deep structures as compared to spread-out representations in shallow structures. This indicates that the representations from GCN and ResGCN converge to a narrow subspace with deep networks. Applying our framework (bottom row) addresses this issue, resulting in *spread-out* representations with deeper structures. This suggests that the application of our framework enhances the expressivity of GCNs.

## K   Expressivity Analysis with Deep Network Structures

In Figure 11, we visualize the impact of over-smoothing by plotting node representations learned by GCN, ResGCN, and our method. The t-SNE embeddings of the representations obtained from the last layer of shallow GCN networks ($L = \{2, 4\}$) and deep GCN networks ($L = \{32, 64\}$) are shown. With vanilla GCN, the representations are organized in clusters and spread out in space for shallow networks. However, for deep networks, the representations lose their organization and collapse to a curved-shaped region. In ResGCNs, the cluster organization is maintained in deep structures, however the separation between clusters becomes less distinct. Also, the representations lie close together within a constricted curved-shaped region compared to that with shallow structures. This is in accordance with Lemma 1 and Corollary 1. The application of our framework (bottom row) results in comparatively *spread-out* representations at the shallow structure ($T/L = 4$), which is in accordance with Theorem 2. Furthermore, the representations remain *spread-out* even at deep structures, indicating improved expressivity as stated in Corollary 1. This demonstrates the effectiveness of our framework in enhancing the expressivity of GCNs.

## L   Effect of the Number of Monte Carlo Samples $S$

We investigate how the number of Monte Carlo samples $S$ used to estimate the ELBO in Equation equation 8 affects the performance of our method. The results are reported in Table 11. We find that a higher number of samples i.e. $S = \{3, 5, 10\}$ performs significantly better compared to $S = \{1, 2\}$. The best performance is achieved with $S = 5$. As further shown in the hyperparameter details in sections G.2 and G.2, $S = 5$ consistently yields the best results in nearly all cases, although a few instances perform best with $S = 3$ or

$S = 10$. Based on these findings, we recommend searching for the hyperparameter $S$ in a range centered around 5 when applying our method.

Table 11: Node classification performance of our method on the Cora and Citeseer datasets for different settings of the number of Monte Carlo samples $S$.

| $S$ | Cora | Citeseer |
|---|---|---|
| 1 | 81.76±0.48 | 74.05±0.20 |
| 2 | 82.31±0.17 | 74.20±0.45 |
| 3 | 86.43±0.24 | 76.06±0.33 |
| 5 | 86.83±0.13 | 77.90±0.37 |
| 10 | 86.70±0.60 | 76.95±0.25 |

## M  Performance with Different Settings of $\alpha$ and $\beta$

We evaluate the effect of the hyperparameters $\alpha$ and $\beta$ on the performance of our method, and the results are reported in Table 12. For small-scale datasets (Cora, Citeseer), the setting $\alpha = 5$, $\beta = 2$ yields the best results, while for larger datasets (e.g., ogb-arxiv), a prior with $\alpha = 25$, $\beta = 2$ is optimal. As shown in the hyperparameter details in sections G.2 and G.2, this pattern holds consistently: smaller datasets in Table 1 favor $\alpha = 5$, $\beta = 2$, whereas larger datasets like ogb-arxiv and ogb-proteins perform best with $\alpha \in \{20, 25\}$ and $\beta = 2$. These findings suggest that the recommended search range for $\alpha$ and $\beta$ should be set according to the dataset size.

Table 12: Node classification performance of our method on the Cora, Citeseer, and ogb-arxiv datasets for different settings of the hyperparameters $(\alpha, \beta)$.

| $(\alpha, \beta)$ | Cora | Citeseer | ogb-arxiv |
|---|---|---|---|
| (2, 5) | 80.16±0.53 | 70.63±0.93 | 64.25±0.93 |
| (2, 2) | 86.40±0.51 | 75.50±0.38 | 67.60±0.45 |
| (5, 2) | 86.83±0.13 | 77.90±0.37 | 70.08±0.16 |
| (10, 2) | 85.85±0.15 | 76.36±0.38 | 70.68±0.14 |
| (20,2) | - | - | 71.21±0.56 |
| (25,2) | - | - | 72.79±0.30 |

## N  Performance on Link Prediction Task

We further validate the performance of our framework in link prediction task by comparing it with Res-GCN on the biomedical datasets. The datasets we use are (a) **BioSNAP-DTI**: Drug Target Interaction network with 15,139 drug-target interactions between 5,018 drugs and 2,325 proteins, (b) **BioSNAP-DDI**: Drug-Drug Interactions with 48,514 drug-drug interactions between 1,514 drugs extracted from drug labels and biomedical literature, (c) **HuRI-PPI**: HI-III human PPI network contains 5,604 proteins and 23,322 interactions generated by multiple orthogonal high-throughput yeast two-hybrid screens. and (d) **DisGeNET-GDI**: gene-disease network with 81,746 interactions between 9,413 genes and 10,370 diseases curated from GWAS studies, animal models, and scientific literature. For all interaction datasets, we represent the interactions as binary adjacency matrix **A** with 0 representing the absence of an interaction and 1 representing the presence of an interaction between two entities in a network. We perform a random split of interactions within a biological network, partitioning them into train/validation/test sets following a ratio of $7 : 1 : 2$. For each set, we acquire pairs of nodes that exhibit interactions between them (positive pairs) from the adjacency matrix **A**. In order to form a balanced dataset, we randomly select an equal number of node pairs with no interactions (negative pairs). The results in Table 13 show that applying our framework improves the link prediction performance on both the AUROC and AUPRC metric.

Table 13: Average AUPRC and AUROC for GCN and Ours on biomedical interaction prediction

| Dataset | Method | AUPRC | AUROC |
|---------|--------|-------|-------|
| DTI | ResGCN | $0.896 \pm 0.006$ | $0.914 \pm 0.005$ |
| | Ours+ResGCN | $\mathbf{0.925 \pm 0.002}$ | $\mathbf{0.933 \pm 0.002}$ |
| DDI | ResGCN | $0.961 \pm 0.005$ | $0.962 \pm 0.004$ |
| | Ours+ResGCN | $\mathbf{0.983 \pm 0.002}$ | $\mathbf{0.982 \pm 0.003}$ |
| PPI | ResGCN | $0.894 \pm 0.002$ | $0.907 \pm 0.006$ |
| | Ours+ResGCN | $\mathbf{0.907 \pm 0.003}$ | $\mathbf{0.918 \pm 0.002}$ |
| GDI | ResGCN | $0.909 \pm 0.002$ | $0.906 \pm 0.002$ |
| | Ours+ResGCN | $\mathbf{0.933 \pm 0.001}$ | $\mathbf{0.945 \pm 0.001}$ |

