# OpenReview forum: "Bayesian Neighborhood Adaptation for Graph Neural Networks"
_TMLR — Accepted by TMLR_

### Review · Reviewer_3vY9 · 2025-04-20

**Summary Of Contributions:**

This paper proposes a Bayesian Neighborhood Adaptation (BNA) framework that automatically infers the appropriate neighborhood scope (i.e., number of hops) in Graph Neural Networks (GNNs) using a nonparametric Bayesian prior (Beta-Bernoulli process). By modeling hop-level contribution probabilities and sampling node features with a binary mask, the method adaptively controls the message passing range during training.

**Audience:**

Yes

**Claims And Evidence:**

Yes

**Requested Changes:**

1. Improve motivation by clearly articulating the rationale for using Bayesian inference over other methods and explicitly comparing with alternative approaches (e.g., attention-based dynamic scope, reinforcement learning).

2. Reorganize Section 3 to build narrative coherence between model formulation, inference strategy, and integration into GNNs.

3. At the beginning of Section 5, introduce a roadmap with subsections for each experiment to improve clarity.

4. Establish stronger connectivity between main paper and appendix, with better notation alignment and referencing.

5. Provide more nuanced insights into when the increased complexity of this method is justified, especially compared to lightweight methods like GPR-GCN or MixHop.

6. Incorporate recent GCN advancements in related work
The current categorization of connection-based methods and GCN improvements is incomplete.
- In particular, the authors should discuss:
  - Cooperative Graph Neural Networks (ICML 2024), which introduce task-aware cooperative message passing strategies and dynamic connection weighting.
  - Learning from the Dark: boosting graph convolutional neural networks with diverse negative samples. (AAAI 2022), which enhances GCNs by learning from diverse and informative negative samples.
  - Layer-Diverse Negative Sampling (TMLR 2024), which improves expressivity and depth robustness by diversifying supervision across GNN layers.

7. Thoroughly revise the manuscript for grammatical clarity, sentence structure, and typographic consistency, such as under Eq. (4):  "O" is the is the dimensionality of the feature vector. "O" should be "W_l"?

**Strengths And Weaknesses:**

**Strengths**
1. Timely focus on a known issue: The paper addresses the critical issue of choosing the proper neighborhood scope in GNNs, which impacts both performance and computational cost.

2. Comprehensive experiments: The authors evaluate their method across a wide range of GNN backbones and datasets, and include expressivity analysis, uncertainty calibration, scalability, and biomolecular graph case studies.

3. Theoretical contribution: The paper includes non-trivial expressivity analysis, extending previous work (e.g., Oono & Suzuki, 2019) and showing how the proposed method avoids feature collapse.

4. Framework generality: The method is applicable to multiple backbone models, such as GCN, ResGCN, GCNII, and GAT.

**Weaknesses and Concerns**
1. **Motivation and Conceptual Framing**
- **Unconvincing motivation for using Bayesian inference.** The paper proposes a Bayesian approach to infer neighborhood scope, but it does
 not clearly justify:
  - Why Bayesian modelling is particularly suited to this task over other variational or heuristic adaptive methods.
  - What fundamental advantagedo  Beta-Bernoulli processes offer in this context over simpler stochastic modeling techniques.

- **Weak taxonomy of related work.** The literature review in Section 2.3 on Bayesian GNNs lacks a clear categorisation or comparison. The paper loosely groups prior work but does not articulate:
  - Why existing Bayesian GNN methods (e.g., Bayesian-GCNN, G3NN) are inadequate for adaptive neighborhood inference.
  - How this work extends or deviates from existing Bayesian inference practices in GNNs.

2. **Organization and Writing.**
- Disjointed Section 3. The subsections of Section 3 (Bayesian modeling, likelihood, inference) read as disconnected modules. The logical flow should be improved by:
  - Starting with an intuitive explanation and motivating example.
  - Explicitly connecting each subsection and clearly stating design decisions (e.g., why using Concrete Bernoulli).
  - Summarizing the full generative process more clearly.

- Unstructured Section 5. The experimental section covers many topics (expressivity, calibration, scalability, ablation, PPI case study), but lacks a clear mapping between objectives and subsections. The introduction of Section 5 should explicitly outline each experimental goal and assign each to a numbered subsection.

- Appendix and Main Text Disconnect. Several critical derivations and definitions (e.g., inference procedure, model integration details) are deferred to the appendix without adequate referencing or notation consistency. The authors should:
  - Clearly reference which parts of the appendix relate to specific claims in the main text.
  - Use consistent notation across the main paper and appendix (e.g., θ, 𝜋_𝑙, 𝑍).

3 **Methodological and Writing Issues**
- Model complexity vs. gain trade-off is not clearly discussed. The Bayesian modelling and variational inference introduce computational overhead. While training time is reported, the paper lacks:
  - Discussion of when this overhead is justified.
  - Analysis of convergence behaviour and sample efficiency.

- Writing requires polishing. There are frequent grammar and syntax issues (e.g., "a daunting and time-consuming task and tend to be biased", "feature expression in a greater number of dimensions"), which occasionally obscure the technical clarity. The writing should be revised for conciseness and precision.

- Missing discussion on the broader impacts of Bayesian inference. The paper could benefit from a short discussion about when and why uncertainty-aware neighbourhood inference (vs. fixed-depth or deterministic schemes) is especially valuable—e.g., in safety-critical systems, few-shot learning, or adversarial robustness.

---

### Review · Reviewer_QL3K · 2025-05-01

**Summary Of Contributions:**

This paper introduced a stochastic process into the message passing process of GNNs by treating the number of hops as a beta process, which allows one to infer the neighborhood scope of the graph while learning their representations. The paper formulated this process and provided theoretical justification for the design, showing that the introduced stochastic process can make GNNs more expressive. The authors later conducted experimental validation of the model, which includes (1) experimental demonstration of the mechanism of the proposed neighborhood adaptation method and its expressiveness; (2) experimental performance by adding the neighborhood adaptation method as a plug in components on top of existing architectures; (3) ablations and complexity analysis; (4) a case study on biomolecular graph and the interpretability results.

**Audience:**

Yes

**Claims And Evidence:**

Yes

**Requested Changes:**

1. The authors should provide additional explanations on section 5.4 (Uncertainty quantification), showing why and how the proposed method is better than other methods, outside of only numerical results.
2. The authors should provide additional explanations on section 5.8, detailing why the results show that “proteins that are three hops apart are likely to interact”.
3. Ideally, if the authors can quantify or provide theoretical justification of the adaptation method by formulating the role of this neighborhood adaptation method in the regularization process, the theoretical results could be a lot stronger.

**Strengths And Weaknesses:**

### Strengths

1. The proposed method is theoretically supported.

2. Figure 4 experimental results on network expressivity are interesting and convincing
- It seems like a clear demonstration of when the proposed adaptation method is useful (prevent performance collapsing with deep GNN structures). I suggest the authors to analyze deeper into this specific case.

### Weaknesses

1. Lack of intuition
- It does not seem like the authors explained their specific choice of stochastic process. Why beta distribution? The choice of beta distribution reminds me of the Mixup augmentation. Given the ablation experiments when comparing against dropout regularization, the authors should explain how much of the proposed adaptation trick is contributing to the final performance by adding more nuanced/advanced regularization; and how much is contributed by actually treating the message passing behaviour as stochastic process.

2. Experimental performance seems weak.
- Performance improvement introduced from the proposed method seems marginal.

3. Uncertainty quantification results lack explanations.
- The authors argue that the proposed approach enhances uncertainty calibration, giving comparable results to GNN deep ensemble methods. However, this conclusion seems to contradict with Table 1 results, where the standard deviation shown by the proposed method in Table 1 are decently high when comparing against other methods.

4. Biomolecular network results in section 5.8 are confusing
- It is unclear how the results support the conclusion that “proteins that are three hops apart are likely to interact”. If that is the case, should not Figure 7 show a jump on the three hop position?

---

### Review · Reviewer_wY2f · 2025-05-05

**Summary Of Contributions:**

This paper introduces a Bayesian Neighborhood Adaptation (BNA) framework for GNNs that dynamically determines the neighborhood size during training, rather than relying on a fixed hop count or exhaustive hyperparameter search. The core idea is to treat the neighborhood expansion as a stochastic process: a beta process prior is placed over an infinite sequence of possible hop counts. Besides, a conjugate Bernoulli process is used to sample feature-wise masks for each hop, determining which dimensions of the node features from that hop are included in aggregation. The authors claim this approach automatically adapts the neighborhood scope for message passing while jointly learning the GNN parameters.

**Audience:**

Yes

**Claims And Evidence:**

Yes

**Requested Changes:**

See Weaknesses.

**Strengths And Weaknesses:**

Strengths:

The proposed BNA is novel. Different from fixed-depth message passing, BNA-GNN can dynamically learn the size of the receptive field by being jointly trained.

Weakness:

1. The reasons/intuitions for applying the Bernoulli process at the feature level are underexplained. A clearer motivation would help readers understand the design choices.

2. The hyperparameter sensitivity of the proposed methods to the choice of $\alpha$ and $\beta$ is not discussed. Since both parameters control the final size of the receptive field, it would be helpful to discuss how $\alpha,\beta$ were set in experiments, like were they fixed across all runs or treated as tunable parameters.

3. For the link prediction task, the authors can further validate BNA's improvement over GNNs on PPI datasets like OGBL-PPA[1]. This dataset is much larger and denser compared to other benchmarks, which can further examine the efficiency and effectiveness of the proposed method.

[1] Hu, Weihua, et al. "Open graph benchmark: Datasets for machine learning on graphs."

---

### Review · Reviewer_ajXi · 2025-05-05

**Summary Of Contributions:**

Apologies for the late review.

This paper offers Bayesian Neighborhood Adaptation (BNA) for adaptively determining the receptive field for residual GCNs. Layer contributions are modeled as a beta process, and node features are masked according to a Bernoulli prior trained via variational inference. Compared to other methods that rely on empirical heuristics, BNA offers an approach to learn optimal layer and feature contributions.

**Audience:**

Yes

**Claims And Evidence:**

Yes

**Requested Changes:**

Please see weaknesses.

**Strengths And Weaknesses:**

**Strengths**

- The paper is generally well-presented and easy to follow.
- The central ideas of this paper are relatively straightforward and novel.
- All theoretical results are well-argued with relatively straighforward proofs.
- BNA is very effective at mitigating the "oversmoothing" problem of deep message-passing networks.

**Weaknesses**

- The intuition behind why the Beta distribution is used to model layer contributions is underexplained. Motivation for the proposed approach is insufficient.
- The performance benefits conferred by BNA are inconsistent for certain combinations of datasets and models both for node classification and uncertainty calibration.
- There is no discussion of the impact of the Bernoulli process over node features on performance. It would be interesting to see how each probabilistic component of BNA contributes to performance.
- Graph rewiring methods like DropEdge are mentioned in related work, and it's revealed in the appendix that it was used by default in the homophilic datasets, but this is not made explicit in the main text. This raises concerns that BNA relies on competing regularization techniques to acheive good performance.

---

### Decision · Action_Editor_wvPx · 2025-06-18

**Recommendation:** Accept with minor revision

**Additional Comments:**

1. Clarify the Motivation for Bayesian Inference: The authors should provide a clearer rationale for using Bayesian inference over other adaptive methods.
2. Enhance Discussion on Model Complexity: This should include insights into the trade-off between model complexity and performance gains.
3. Polish Writing: The manuscript should be thoroughly revised for grammatical clarity, sentence structure, and typographic consistency. Specificly, section 3, and section 5 should be polished carefully.

Please refer to the reviewers' comments for more details.

**Audience:**

Yes

**Audience Explanation:**

The work addresses a significant issue in Graph Neural Networks (GNNs) related to neighborhood scope adaptation, which is relevant to researchers working on graph machine learning. The findings are likely to be of interest to TMLR's audience.

**Claims And Evidence:**

Yes

**Claims Explanation:**

The reviewers generally agree that the claims made in the submission are supported by accurate and clear evidence. The authors have provided theoretical analysis, experimental results, and ablation studies that validate their proposed Bayesian Neighborhood Adaptation (BNA) framework.